# LEARNING WHEN TO BE UNCERTAIN: A POST-HOC META-MODEL FOR GUIDED UNCERTAINTY LEARNING

## ABSTRACT

Reliable uncertainty quantification remains a major obstacle to the deployment of deep learning models under distributional shift. Existing post-hoc approaches that retrofit pretrained models either inherit misplaced confidence or merely reshape predictions, without teaching the model when to be uncertain. We introduce GUIDE, a lightweight evidential learning meta-model approach that attaches to a frozen deep learning model and explicitly learns how and when to be uncertain. GUIDE identifies salient internal features via a calibration stage, and then employs these features to construct a noise-driven curriculum that teaches the model how and when to express uncertainty. GUIDE requires no retraining, no architectural modifications, and no manual intermediate-layer selection to the base deep learning model, thus ensuring broad applicability and minimal user intervention. The resulting model avoids distilling overconfidence from the base model, improves out-of-distribution detection ($\approx 77\%$) and adversarial attack detection ($\approx 80\%$), while preserving in-distribution performance. Across diverse benchmarks, GUIDE consistently outperforms state-of-the-art approaches, evidencing the need for actively guiding uncertainty to close the gap between predictive confidence and reliability.

## 1 INTRODUCTION

Recent advances in artificial intelligence (AI), particularly in medical imaging Li et al. (2020) and autonomous agents Rudin et al. (2022), have enabled the deployment of deep learning models into numerous real-world applications. In high-stakes domains, however, such as healthcare Miotto et al. (2018) and human-in-the-loop robotics Retzlaff et al. (2024), the reliability and trustworthiness of these models remain a paramount concern. In such contexts, a model must not only recognise the limitations of its predictions but also produce well-calibrated estimates of predictive uncertainty. This requirement is particularly critical in the real world, where distributional shift from out-of-distribution (OOD) inputs and attacks from adversarial inputs, attempting to mislead the model, are prevalent.

Uncertainty quantification (UQ) has been widely studied as a framework for mitigating these reliability challenges He & Jiang (2023). Epistemic uncertainty, stemming from limited or biased data, is commonly modelled using Bayesian neural networks Goan & Fookes (2020), variational inference Blei et al. (2017), or Laplace approximations Fortuin (2022), though these approaches are computationally intensive. More scalable alternatives, including Monte Carlo Dropout Gal & Ghahramani (2016), deep ensembles Lakshminarayanan et al. (2017), adversarial perturbations Schweighofer et al. (2023), and distance-aware models Liu et al. (2020); Van Amersfoort et al. (2020), balance efficiency and expressiveness. Aleatoric uncertainty is typically captured through input-dependent predictors Kendall & Gal (2017), prediction intervals Khosravi et al. (2010), or generative models Kingma et al. (2013). More recently, post-hoc approaches that act directly on pretrained networks have gained traction, including evidential deep learning (EDL) Sensoy et al. (2018) and adaptable BNNs Franchi et al. (2024), although their applicability and scalability remain limited. Recent focus on non-intrusive methods like Whitebox Chen et al. (2019) and EMM Shen et al. (2023), albeit computationally efficient, often degrade in performance under OOD or adversarial conditions.

We introduce the **G**radual **U**ncertainty Refinement via Noise-**D**riven Curriculum (GUIDE) meta-model, a non-intrusive post-hoc evidential meta-model approach that operates on top of a pretrained base model and explicitly learns how and when to be uncertain. GUIDE's two-stage approach (Figure 1) initially performs saliency calibration to determine the (frozen) pretrained model's salient

Figure 1: Overview of the GUIDE meta-model approach, showing the Saliency Calibration and Uncertainty Guided Training stages. GUIDE extracts salient features and weight maps from a pretrained model in a fully post-hoc manner, then generates a noise-driven curriculum to teach the meta-model when to be uncertain. In the figure, $g_\ell$ are evidential projection branches from salient layers, and $u$ is the predicted uncertainty from the Dirichlet evidence.

intermediate features. Then, it leverages this knowledge to create a gradual noise-driven curriculum to teach the meta-model when to be uncertain and to what magnitude. GUIDE retains (and in some cases improves, for example, +16% ID coverage on the CIFAR10 → SVHN dataset pairing) the in-distribution (ID) predictive performance while significantly improving the OOD and adversarial attack robustness. We conduct extensive experiments across various ID/OOD datasets, near- and far-OOD scenarios, and both gradient- and non-gradient-based attacks at multiple perturbation strengths. GUIDE achieves state-of-the-art results in our experimental evaluation, achieving a significant drop in OOD and adversarial predictions compared to state-of-the-art intrusive and non-intrusive post-hoc UQ methods.

Our key contributions are: (1) GUIDE is the first fully post-hoc meta-model approach that explicitly learns when and how to be uncertain by leveraging saliency calibration and noise-driven curriculum learning; (2) theoretical guarantees for GUIDE's soundness and convergence; and (3) extensive experiments demonstrating that GUIDE retains informative structure from the pretrained model while achieving state-of-the-art robustness to OOD and adversarial inputs compared to both intrusive and non-intrusive post-hoc baselines. To the best of our knowledge, GUIDE is the first non-intrusive, fully post-hoc model that reliably estimates predictive uncertainty and explicitly teaches the model when and how to be uncertain within its predictions.

## 2 RELATED WORK

**Uncertainty Quantification.** Uncertainty quantification (UQ) is essential for reliable deep learning, especially in safety-critical settings He & Jiang (2023). Epistemic uncertainty, due to limited or biased training data, is typically addressed with Bayesian neural networks Goan & Fookes (2020), variational inference Blei et al. (2017), or Laplace approximations Fortuin (2022), though at high cost. More scalable methods, including Monte Carlo Dropout Gal & Ghahramani (2016), deep ensembles Lakshminarayanan et al. (2017), adversarial perturbations Schweighofer et al. (2023), and distance-aware models Liu et al. (2020); Van Amersfoort et al. (2020), trade efficiency for expressiveness. In contrast, aleatoric uncertainty, arising from data noise, is typically managed with input-dependent predictors Kendall & Gal (2017); Guo et al. (2017), prediction intervals Khosravi et al. (2010); Tagasovska & Lopez-Paz (2019), or generative models Kingma et al. (2013); Goodfellow et al. (2014), while test-time augmentation Ayhan & Berens (2018) is simple but type-agnostic. Since both forms often coexist He & Jiang (2023), hybrid methods such as ensembles with prediction intervals Pearce et al. (2018) or conformal prediction Angelopoulos et al. (2023) aim to capture both uncertainty types, albeit with added cost or calibration overheads.

**Post-hoc Methods.** Most of the previous approaches require modifying or retraining the base model, limiting their applicability in large-scale or black-box settings. To overcome this, recent work introduced post-hoc uncertainty quantification approaches that act directly on pretrained deterministic models. Notable examples include Monte Carlo Dropout Gal & Ghahramani (2016) and intrusive

evidential extensions Sensoy et al. (2018); Franchi et al. (2024), which attach uncertainty estimation layers and perform light fine-tuning on top of the pretrained backbone.

**Non-Intrusive Post-hoc Methods.** Recent work avoids retraining or modifying the base model by devising auxiliary models on frozen pretrained representations. Representative examples include Whitebox Chen et al. (2019) and EMM Shen et al. (2023), which attach lightweight uncertainty estimators to intermediate or final pretrained features. These non-intrusive methods are broadly applicable and efficient, suitable even in black-box settings without training pipelines. Their efficiency, however, often yields a robustness cost. Their uncertainty estimates are brittle under out-of-distribution (OOD) or adversarial inputs, frequently assigning unwarranted confidence to harmful samples.

GUIDE, unlike existing UQ research, is a fully post-hoc non-intrusive approach that not only produces effective predictive uncertainty estimates but also explicitly teaches the model when and how to be uncertain. GUIDE's design yields significantly improved robustness to both OOD shifts and adversarial attacks, distinguishing GUIDE from prior UQ approaches.

## 3 PRELIMINARIES

We study the standard supervised classification setting, where the objective is to train a predictive model over a finite labelled set. The input domain is $X \subseteq \mathbb{R}^d$ and the output space is $Y = \{1, \ldots, K\}$, corresponding to $K$ discrete classes. The training (ID) data $\mathcal{D} = \{(x_i, y_i)\}_{i=1}^N$ comprises samples $(x_i, y_i) \in X \times Y$ drawn independently and identically from the joint distribution $p(x, y) = p(x)\, p(y \mid x)$. The central task is to estimate the conditional distribution $p(y \mid x)$.

Beyond accuracy, we focus on assessing predictive uncertainty. A pretrained deterministic model $f_\theta$ trained on ID data, though often accurate, is typically overconfident, especially on OOD and adversarial inputs. This occurs since its softmax outputs $\sigma(f_\theta(x))$ combined with cross-entropy loss promote high confidence even for out-of-distribution samples. GUIDE reduces such overconfidence in a fully post-hoc manner, using only the base model's outputs and features without modifying $f$ or $\theta$, as intrusive changes may be inaccessible, infeasible for large models, or harmful to performance.

## 4 GUIDE

Our **G**radual **U**ncertainty Ref**i**nement via Noise-**D**riv**e**n Curriculum (GUIDE) meta-model approach (Figure 1) enhances uncertainty calibration and robustness to OOD and adversarial inputs. GUIDE adaptively identifies both salient intermediate layers of the pretrained model to connect them to evidential linear layers and salient weight maps. These weight maps are then exploited to construct a monotonic noise-driven curriculum with progressively perturbed inputs. This curriculum enables GUIDE to learn calibrated confidence estimates, thus mitigating the propagation of overconfidence typically inherited from the pretrained model and improving robustness under distributional shift.

### 4.1 SALIENCY CALIBRATION

The saliency calibration stage of GUIDE identifies salient components at both the layer and instance levels of the pretrained model. Given the logits of a forward pass $z = f_\theta(x)$ and one-hot vector $e_y$, we initialise the relevance at the output as $\mathcal{R}_L(x) = z\, e_y$. Employing the LRP-$\epsilon$ rule Montavon et al. (2019), we propagate relevance to each hidden layer of the pretrained model:

$$\mathcal{R}_{\ell-1}(x) = \Big( \frac{a_{\ell-1} W_\ell^\top}{\langle a_{\ell-1}, W_\ell^\top \rangle + \epsilon} \Big) \odot \mathcal{R}_\ell(x), \qquad \ell = L, L-1, \ldots, 1 \tag{1}$$

where $a_{\ell-1}$ are the activations, $W_\ell$ are the weights, and $\epsilon > 0$ is a small stabilisation term that partially absorbs relevance when there is a contradiction between consecutive layers[1].

This formulation yields relevance maps $\{\mathcal{R}_\ell(x)\}_{\ell=1}^L$ describing how each layer contributes to the final prediction. Each $R_\ell(x)$ is a vector of size $\dim(\ell)$ with each element signifying the relevance

---

[1]equation 1 is expressed in vectorised notation for compactness and should be interpreted as an abbreviation of the per-neuron redistribution rule, consistent with prior work.

attributed to each neuron of the $\ell$-th layer. This propagation rule preserves class-relevant evidence and is applicable across standard deep learning components (e.g., convolutional, dense, pooling). We then quantify each hidden layer's global importance by computing the average relevance magnitude over the $N$ input samples:

$$M_\ell = \frac{1}{N} \sum_{i=1}^{N} \frac{||\mathcal{R}_\ell(x_i)||_1}{|\mathcal{R}_\ell(x_i)|} \tag{2}$$

which enables the selection of hidden layers that hold sufficient information and can be used to train the evidential meta-model. Specifically, sorting layers by $M_\ell$, we select the smallest subset $L_{\text{sal}}$ that covers at least a fraction $\eta$ of the total relevance mass:

$$\sum_{\ell \in L_{\text{sal}}} M_\ell \geq \eta \sum_\ell M_\ell \tag{3}$$

where $\eta \in (0, 1]$ is the cumulative relevance coverage threshold. For instance, $\eta = 0.9$ selects the set of layers that collectively account for 90% of relevance. This principled criterion avoids arbitrary layer selection Shen et al. (2023) and ensures that the evidential meta-model operates on semantically informative representations.

**Theorem 1.** *Let $\eta \in (0, 1]$ denote the saliency coverage threshold, $\kappa \in (0, 1]$ the alignment of saliency scores with the Fisher information mass, and $\rho \in (0, 1]$ the information preserved by the projection. Under local linearisation and restricting to cross-depth correlated blocks, GUIDE's saliency calibration retains at least $\rho\kappa\eta$ of the pretrained Fisher trace with respect to $\beta$. Moreover, if the saliency normalisation is stable and the projections are well-conditioned, then $\kappa \approx 1$ and $\rho \approx 1$, so that the retained Fisher fraction is essentially determined by $\eta$.*

The corresponding proof is available in Appendix A. To construct a noise-driven curriculum aligned with salient input features, we define per-input weight maps from the input-level relevance $\mathcal{R}_0(x)$:

$$\mathcal{W}(x)[h, w] = \frac{|\sum_{c=1}^{C} \mathcal{R}_0(x)[h, w, c]|}{\max_{h', w'} |\sum_{c=1}^{C} \mathcal{R}_0(x)[h', w', c]|} \tag{4}$$

Here, $C$ denotes channels and $(h, w)$ spatial indices. The normalisation yields a weight map where higher values correspond to stronger contributions to the model's prediction Bach et al. (2015), and which can be used to prioritise input feature perturbations.

Both global layer relevance scores $M_\ell$ and input-level weight maps $\mathcal{W}(x)$ are obtained in a single backward pass: as relevance propagates from output to input, we accumulate per-layer relevance for $M_\ell$ and retain $\mathcal{R}_0(x)$ for $\mathcal{W}(x)$. This reuse adds no extra forward or backward passes beyond standard relevance propagation Montavon et al. (2019).

## 4.2 UNCERTAINTY GUIDED TRAINING

The uncertainty-guided training stage of GUIDE constructs and trains a small Dirichlet-based meta-model $g_\phi$ exclusively on features extracted from the selected layers $L_{\text{sal}}$ and guides it on when to be uncertain using a monotonic soft-target curriculum. For each selected layer $\ell \in L_{\text{sal}}$, the feature map is flattened $\Phi_\ell(x)$ and projected to $\mathbb{R}^K$ using a branch of multiple linear layers $g_\ell$, such that $g_\ell(\Phi_\ell(x)) \in \mathbb{R}^K$. The resulting meta-features from all selected layers, $\{g_\ell(\Phi_\ell(x))\}_{\ell \in L_{\text{sal}}}$, are concatenated into a single vector $\Psi(x)$ and passed through a final linear head $g_{\text{out}}$, yielding $\alpha(x) = \exp(g_{\text{out}}(\Psi(x))) + 1$. Here $\alpha(x)$ parametrises a log-concentrated Dirichlet distribution $\text{Dir}(\pi|\alpha)$ over the categorical label probabilities $\pi = [\pi_1, \dots, \pi_K] \in \Delta^{K-1}$, where an additive 1 enforces strictly positive parameters as standard in evidential deep learning Sensoy et al. (2018). This strategy enables the meta-model to represent both predictions and epistemic uncertainty. High total evidence $S = \sum_k \alpha_k$ implies confident predictions, while low $S$ reflects uncertainty. All $\phi$ parameters are learned from scratch, while the pretrained model's parameters $\theta$ are frozen throughout.

To induce robustness and *teach* the meta-model when to be uncertain, we create a targeted noise-driven curriculum. Firstly, a monotonic exponential schedule is constructed $s_t = 1 - e^{-\gamma t}$ for $t = 0, \dots, T$, where $s_t \in [0, 1]$ is the target fraction of noise corrupted pixels at stage $t \in T$, and $\gamma > 0$ is the rate of noise corruption. The first image ($t = 0$) represents the clean view; then, the exponential form ensures fine-grained corruption at early stages (where decision boundaries are sensitive) and coarser granularity at higher noise levels.

To ensure salient features are targeted first, we leverage the weight maps $\mathcal{W}(x)$ from the saliency calibration stage. A global saliency budget $\tilde{\mathcal{W}} = \frac{1}{HW} \sum_{h,w} \mathcal{W}(x)[h,w]$ is defined as to construct a per-pixel corruption probability $p_t(h,w)$ satisfying two conditions: the expected global corruption matches the budget $s_t$, and pixel-wise probabilities are monotonic in $s_t$. It is defined as:

$$p_t(h,w) = \begin{cases} \frac{s_t}{\tilde{\mathcal{W}}} \cdot \mathcal{W}(x)[h,w], & s_t \leq \tilde{\mathcal{W}} \\ \mathcal{W}(x)[h,w] + \frac{s_t - \tilde{\mathcal{W}}}{1 - \tilde{\mathcal{W}}} \cdot (1 - \mathcal{W}(x)[h,w]), & s_t > \tilde{\mathcal{W}} \end{cases} \tag{5}$$

This ensures that low-noise perturbations are focused on high-saliency regions, while high-noise settings affect the entire image. To actually apply the noise, we sample a single base mask $m \sim U[0,1]^{H \times W}$, reused across all $t$ for a given data point. For each stage $t$, we corrupt $x$ to obtain $\tilde{x}_t$:

$$\tilde{x}[h,w] = \begin{cases} 0, & m[h,w] < p_t(h,w)/2, \\ 1, & m[h,w] > 1 - p_t(h,w)/2, \\ x[h,w], & \text{otherwise.} \end{cases} \tag{6}$$

This procedure creates stochastic binary corruption, preserving the expected noise budget per image and ensuring that $\tilde{x}_{t+1}$ is never less noisy than $\tilde{x}_t$.

To teach the meta-model when to express uncertainty, we construct soft targets that respond to both input corruption and the model's confidence, as we want the model to be confident for the clean input ($t = 0$), and very uncertain for the noisiest input ($t = T$). For each corrupted view, the base model's predicted confidence in the true class $c_t = \langle \sigma(f_\theta(\tilde{x}_t)), y \rangle$ is used along with the noise level to define an uncertainty target $\tilde{s}_t = s_t \cdot (1 - \frac{1}{2}c_t^2)$. The soft target for the meta-model is a convex combination of the uniform distribution and the true label $\tilde{y}_t = \frac{\tilde{s}_t}{K} \cdot \mathbf{1}_K + (1 - \tilde{s}_t) \cdot y$. This encourages high certainty for confident, low-noise inputs, and near-uniform predictions for highly corrupted or uncertain examples. The soft targets remain monotonic in $t$ in expectation, ensuring consistency in the learned uncertainty behaviour. Next, we generate a curriculum over corruption strengths. Instead of training the meta-model on all strengths uniformly, we define an epoch-dependent difficulty index $\rho_e = \left(\frac{e}{E-1}\right)^2$ for every epoch $e \in E$. At each epoch $e$, the sampling distribution over the discrete noise level $\{s_0, \ldots, s_T\}$ is given by $\kappa_e(s) \propto (1 - \rho_e)(1 - s) + \rho_e \cdot s$ and determines which corruption levels populate the training batches.. In early epochs ($\rho_e \approx 0$) the sampling distribution concentrates mass on small $s$, so a higher proportion of clean or mildly corrupted views are trained upon. In late epochs ($\rho_e \to 1$), the distribution shifts towards large $s$, evidently, strongly corrupted views dominate the training data. Thus, we generate a curriculum that forces a monotone progression of views and soft targets, from clean to fully corrupted, in which we can guide the meta-model on how and when to be uncertain.

Finally, the meta-model is trained using an uncertainty regularised evidence lower bound (ELBO) combined with a self-rejecting evidence (SRE) penalty to discourage overconfident predictions when the predictive mean disagrees with the soft target. For a mini-batch $\mathcal{B}$, the loss is defined as:

$$\mathcal{L} = \frac{1}{|\mathcal{B}|} \sum_{(\tilde{x},\tilde{y}) \in \mathcal{B}} \left[ -\sum_{k=1}^{K} \tilde{y}_k (\psi(\alpha_k) - \psi(S)) + \lambda_{\text{kl}} \cdot \text{KL}(\text{Dir}(\alpha)||\text{Dir}(\beta)) \right] + \underbrace{\frac{S}{K} \cdot (1 - \langle \tilde{y}, \hat{p} \rangle)}_{\text{SRE penalty}} \tag{7}$$

where $\psi(\cdot)$ is the digamma function, $\hat{p}$ is the predictive mean. By weighting the expected log-likelihood by the soft target $\tilde{y}$, it makes evidence sharp for near-one-hot labels, flat/uniform for highly corrupted inputs, and intermediate for moderate noise, giving a graded uncertainty response. The self-rejecting evidence penalty then suppresses any large $S$ that is not supported by target agreement, eliminating misplaced confidence while leaving well-aligned, high-evidence predictions untouched. Combined, these two terms guide the meta-model to be confident only when justified and gradually/monotonically uncertain everywhere else, tightening the gap between in- and out-of-distribution behaviour.

## 5 EVALUATION

We assess the performance of GUIDE through an extensive set of experiments, benchmarking it against state-of-the-art post-hoc UQ methods across 10 independent trials. The evaluation examines

Table 1: Mean accuracy, OOD detection, and adversarial attack detection performance with 95% CI of the comparative approaches with a variety of ID and OOD datasets in order of dataset difficulty. The adversarial attack is an L2PGD attack [3] Highlighted cells denote the best performance for each metric. * indicates datasets classed as Near-OOD.

| Type | Pretrained | ABNN | EDL-Head | Whitebox | EMM | EMM + cal | EMM + curric | GUIDE |
|---|---|---|---|---|---|---|---|---|
| | - | Intrusive | Intrusive | Post-hoc | Post-hoc | Post-hoc | Post-hoc | Post-hoc |
| **MNIST → FashionMNIST** | | | | | | | | |
| ID Acc ↑ | 99.80 [99.75, 99.85]% | 99.78 [99.74, 99.82]% | 99.43 [99.31, 99.55]% | 99.92 [99.83, 100.00]% | 99.55 [99.49, 99.60]% | 99.69 [99.62, 99.75]% | 99.21 [99.04, 99.38]% | 99.87 [99.82, 99.92]% |
| ID Cov ↑ | 79.26 [76.31, 82.22]% | 74.88 [69.46, 80.29]% | 77.44 [75.24, 79.65]% | 76.38 [69.77, 83.00]% | 90.58 [87.85, 93.33]% | 92.32 [91.26, 93.38]% | 79.46 [72.51, 86.42]% | 87.05 [85.24, 88.85]% |
| OOD Cov ↓ | 24.89 [18.30, 31.47]% | 18.23 [12.40, 24.07]% | 22.51 [18.78, 26.24]% | 13.50 [10.48, 16.52]% | 31.87 [14.07, 49.68]% | 35.69 [23.92, 47.47]% | 40.18 [30.49, 49.87]% | 7.34 [3.54, 11.14]% |
| Adv Cov ↓ | 25.84 [20.52, 31.16]% | 24.35 [17.79, 30.92]% | 13.86 [11.91, 15.80]% | 9.04 [4.36, 13.72]% | 22.18 [6.91, 37.44]% | 55.47 [46.03, 64.91]% | 62.11 [51.81, 72.41]% | 4.60 [1.61, 7.58]% |
| AUROC ↑ | 84.18 [80.16, 88.21]% | 85.93 [84.50, 87.36]% | 83.23 [80.59, 85.86]% | 88.38 [86.22, 90.54]% | 77.68 [63.70, 91.66]% | 74.54 [64.69, 84.38]% | 72.85 [67.40, 78.30]% | 94.85 [93.44, 96.26]% |
| Adv AUROC ↑ | 83.67 [81.38, 85.95]% | 81.81 [79.13, 84.48]% | 88.73 [86.94, 90.53]% | 90.26 [89.61, 90.92]% | 83.43 [71.25, 95.62]% | 53.56 [44.58, 62.54]% | 53.62 [46.89, 60.36]% | 95.72 [94.71, 96.74]% |
| **MNIST → KMNIST** | | | | | | | | |
| ID Acc ↑ | 99.81 [99.77, 99.85]% | 99.76 [99.72, 99.79]% | 99.57 [99.50, 99.64]% | 99.87 [99.83, 99.92]% | 99.51 [99.43, 99.59]% | 99.56 [99.49, 99.63]% | 99.15 [99.01, 99.29]% | 99.75 [99.66, 99.84]% |
| ID Cov ↑ | 80.69 [79.02, 82.37]% | 85.91 [84.78, 87.04]% | 75.03 [73.06, 77.00]% | 85.93 [84.99, 86.88]% | 92.48 [91.76, 93.21]% | 94.49 [93.57, 95.42]% | 86.67 [83.89, 89.44]% | 89.80 [88.58, 91.02]% |
| OOD Cov ↓ | 18.23 [16.78, 19.68]% | 22.90 [21.13, 24.66]% | 18.54 [15.62, 21.47]% | 8.63 [6.84, 10.42]% | 18.47 [13.48, 23.45]% | 12.89 [8.29, 16.10]% | 32.31 [29.27, 35.35]% | 9.00 [6.04, 11.95]% |
| Adv Cov ↓ | 38.35 [36.06, 40.63]% | 46.27 [43.72, 48.82]% | 17.78 [14.75, 20.82]% | 9.60 [6.76, 10.43]% | 50.97 [34.01, 67.92]% | 49.89 [38.62, 61.15]% | 67.51 [64.26, 70.76]% | 9.00 [6.04, 11.95]% |
| AUROC ↑ | 88.27 [87.63, 88.90]% | 86.79 [85.89, 87.69]% | 85.38 [83.97, 86.79]% | 94.64 [93.91, 95.37]% | 89.67 [86.32, 93.02]% | 94.01 [91.63, 96.50]% | 82.62 [80.75, 84.50]% | 96.68 [96.19, 97.16]% |
| Adv AUROC ↑ | 77.20 [76.33, 78.08]% | 74.87 [73.40, 76.35]% | 85.63 [83.91, 87.34]% | 94.42 [93.74, 95.10]% | 56.29 [39.87, 72.71]% | 61.42 [51.46, 71.37]% | 52.33 [48.74, 55.92]% | 95.80 [94.87, 96.73]% |
| **MNIST → EMNIST*** | | | | | | | | |
| ID Acc ↑ | 99.80 [99.77, 99.82]% | 99.71 [99.67, 99.75]% | 99.49 [99.43, 99.55]% | 99.89 [99.84, 99.94]% | 99.61 [99.54, 99.68]% | 99.65 [99.55, 99.74]% | 99.01 [98.87, 99.15]% | 99.89 [99.83, 99.94]% |
| ID Cov ↑ | 81.64 [80.87, 82.41]% | 86.79 [86.16, 87.42]% | 76.29 [74.04, 78.54]% | 84.10 [82.33, 85.88]% | 89.58 [87.44, 91.71]% | 92.65 [91.51, 93.80]% | 87.59 [84.72, 90.45]% | 84.94 [83.50, 86.38]% |
| OOD Cov ↓ | 24.09 [23.04, 25.15]% | 28.65 [27.26, 30.05]% | 22.98 [20.96, 25.00]% | 17.59 [16.42, 18.76]% | 35.97 [30.27, 41.66]% | 25.80 [21.56, 30.04]% | 41.71 [38.20, 45.21]% | 15.82 [13.15, 18.50]% |
| Adv Cov ↓ | 31.60 [30.56, 32.63]% | 43.40 [41.36, 45.44]% | 17.54 [15.74, 19.34]% | 12.23 [9.88, 14.58]% | 51.21 [37.75, 64.68]% | 42.90 [35.39, 50.40]% | 59.73 [55.46, 64.00]% | 10.05 [5.74, 14.35]% |
| AUROC ↑ | 85.46 [84.80, 86.13]% | 83.57 [82.87, 84.26]% | 83.25 [82.12, 84.38]% | 89.47 [88.72, 90.22]% | 77.37 [72.85, 81.89]% | 84.72 [81.59, 87.85]% | 75.11 [72.21, 78.01]% | 91.08 [89.87, 92.29]% |
| Adv AUROC ↑ | 82.02 [81.28, 82.75]% | 76.47 [75.34, 77.59]% | 86.13 [84.89, 87.37]% | 91.96 [90.81, 93.10]% | 56.72 [43.30, 70.14]% | 65.67 [58.16, 73.18]% | 57.48 [53.74, 61.22]% | 93.76 [92.40, 95.11]% |
| **CIFAR10 → SVHN** | | | | | | | | |
| ID Acc ↑ | 93.27 [91.54, 95.00]% | 96.02 [95.61, 96.43]% | 88.62 [87.10, 90.14]% | 90.14 [88.88, 91.40]% | 76.30 [58.15, 94.45]% | 92.07 [89.64, 94.49]% | 91.56 [88.02, 95.09]% | 88.46 [85.63, 91.29]% |
| ID Cov ↑ | 64.48 [57.10, 71.87]% | 63.94 [61.70, 66.15]% | 10.85 [8.50, 13.21]% | 69.71 [64.85, 74.56]% | 66.52 [46.94, 86.09]% | 60.54 [45.54, 75.54]% | 37.27 [12.33, 62.22]% | 80.65 [70.18, 91.12]% |
| OOD Cov ↓ | 12.03 [8.28, 15.78]% | 10.85 [8.50, 13.21]% | 23.55 [17.56, 29.54]% | 17.89 [13.36, 22.41]% | 16.43 [6.23, 26.63]% | 21.62 [12.72, 30.51]% | 13.70 [1.35, 28.74]% | 19.34 [12.64, 26.04]% |
| Adv Cov ↓ | 66.01 [48.27, 83.76]% | 71.05 [59.25, 82.86]% | 24.78 [18.69, 30.87]% | 37.06 [29.73, 44.40]% | 42.27 [16.30, 68.23]% | 63.74 [53.01, 74.47]% | 39.64 [14.84, 64.43]% | 48.04 [33.87, 62.21]% |
| AUROC ↑ | 81.96 [78.49, 85.43]% | 79.58 [77.38, 81.78]% | 77.57 [70.96, 84.18]% | 81.95 [77.76, 86.14]% | 78.76 [62.48, 95.05]% | 74.86 [66.19, 83.53]% | 56.02 [38.78, 73.25]% | 89.36 [82.97, 95.75]% |
| Adv AUROC ↑ | 57.95 [52.85, 63.06]% | 41.44 [36.35, 46.52]% | 76.78 [69.85, 83.70]% | 71.07 [66.66, 75.49]% | 69.86 [64.12, 75.60]% | 50.97 [37.99, 63.95]% | 42.66 [32.87, 52.44]% | 78.07 [70.59, 85.55]% |
| **CIFAR10 → CIFAR100*** | | | | | | | | |
| ID Acc ↑ | 92.27 [91.26, 93.27]% | 93.48 [92.71, 94.25]% | 87.22 [85.63, 88.82]% | 83.49 [79.71, 87.27]% | 81.09 [62.10, 100.00]% | 89.20 [88.15, 90.24]% | 65.96 [30.38, 100.00]% | 88.90 [87.93, 89.88]% |
| ID Cov ↑ | 51.94 [49.74, 54.13]% | 53.69 [50.69, 56.68]% | 51.69 [46.45, 56.93]% | 56.06 [50.80, 61.32]% | 59.95 [57.63, 62.28]% | 57.42 [51.86, 62.99]% | 50.58 [21.12, 80.04]% | 56.67 [50.56, 62.77]% |
| OOD Cov ↓ | 20.97 [19.22, 22.73]% | 20.17 [17.23, 23.10]% | 28.70 [24.36, 33.05]% | 25.29 [21.18, 29.40]% | 27.05 [23.44, 30.66]% | 24.50 [20.27, 28.72]% | 40.52 [6.79, 74.26]% | 24.01 [19.36, 28.66]% |
| Adv Cov ↓ | 24.79 [21.82, 27.75]% | 28.12 [22.84, 33.39]% | 22.75 [17.50, 28.00]% | 17.17 [13.23, 21.10]% | 27.85 [22.61, 33.08]% | 23.81 [18.63, 28.98]% | 37.41 [4.75, 70.07]% | 17.09 [12.65, 21.52]% |
| AUROC ↑ | 70.41 [69.50, 71.33]% | 71.81 [71.39, 72.23]% | 65.66 [64.37, 66.95]% | 70.09 [67.11, 73.07]% | 71.81 [69.61, 74.01]% | 72.09 [70.65, 73.52]% | 55.55 [50.51, 60.58]% | 71.61 [70.53, 72.68]% |
| Adv AUROC ↑ | 65.24 [64.52, 65.97]% | 64.50 [63.85, 65.14]% | 69.90 [69.11, 70.69]% | 75.96 [72.87, 79.06]% | 70.86 [68.66, 73.06]% | 71.81 [70.90, 72.72]% | 57.89 [53.59, 62.18]% | 76.01 [75.26, 76.76]% |
| **Oxford Flowers (low-shot) → Deep Weeds** | | | | | | | | |
| ID Acc ↑ | 99.14 [98.68, 99.59]% | 98.86 [98.08, 99.63]% | 84.11 [80.50, 87.72]% | 97.22 [93.89, 100.00]% | 97.69 [77.05, 98.33]% | 92.70 [78.23, 100.00]% | 79.66 [70.97, 88.35]% | 90.84 [89.09, 92.59]% |
| ID Cov ↑ | 45.12 [29.87, 60.36]% | 74.24 [66.46, 82.01]% | 57.57 [18.86, 96.28]% | 33.80 [1.92, 65.68]% | 49.10 [31.50, 66.69]% | 62.44 [48.67, 76.21]% | 48.52 [33.10, 63.95]% | 74.98 [68.21, 81.76]% |
| OOD Cov ↓ | 18.95 [2.29, 35.62]% | 19.58 [4.41, 34.76]% | 35.60 [6.38, 77.57]% | 13.62 [5.45, 32.70]% | 10.50 [1.93, 19.06]% | 13.71 [5.78, 21.63]% | 10.49 [5.62, 15.36]% | 19.52 [9.88, 29.16]% |
| Adv Cov ↓ | 7.92 [1.59, 17.42]% | 11.78 [1.26, 22.29]% | 32.33 [11.15, 75.81]% | 9.35 [5.62, 24.32]% | 2.76 [0.10, 5.41]% | 3.71 [1.03, 6.38]% | 3.40 [1.93, 4.87]% | 9.74 [3.51, 15.98]% |
| AUROC ↑ | 66.77 [56.61, 76.94]% | 82.44 [72.13, 92.74]% | 59.52 [38.10, 80.94]% | 53.86 [30.46, 77.26]% | 69.61 [58.45, 80.76]% | 78.72 [72.23, 85.22]% | 66.00 [54.03, 77.97]% | 81.63 [71.25, 92.00]% |
| Adv AUROC ↑ | 75.76 [67.57, 83.95]% | 86.31 [78.65, 93.97]% | 63.55 [43.30, 83.81]% | 57.53 [33.62, 81.45]% | 78.67 [67.18, 90.17]% | 87.62 [82.62, 92.62]% | 73.53 [63.77, 83.29]% | 88.56 [80.84, 96.27]% |

both predictive performance and the quality of uncertainty estimates under OOD shifts and adversarial perturbations. All code and replication materials are publicly available in our open-source repository [2].

**Datasets.** Adopting the procedure on UQ-based evaluation from recent research Shen et al. (2023); Sensoy et al. (2018), we evaluate all approaches on the MNIST LeCun et al. (1998), Fashion-MNIST Xiao et al. (2017), KMNIST Clanuwat et al. (2018), EMNIST Cohen et al. (2017), CIFAR10 Krizhevsky et al. (2009), CIFAR100 Krizhevsky et al. (2009), SVHN Netzer et al. (2011), Oxford Flowers Netzer et al. (2011), and Deep Weeds Olsen et al. (2019) datasets which were selected to cover a diverse set of domains and challenges. In the following experiments, near-OOD datasets are those that share some degree of class overlap with the ID dataset Yang et al. (2022).

**Comparative Approaches.** We compare GUIDE against state-of-the-art post-hoc UQ methods, both intrusive (change the architecture of the pretrained model) and fully post-hoc: a pretrained deterministic network, ABNN Franchi et al. (2024), an evidential head Sensoy et al. (2018), Whitebox Chen et al. (2019), and EMM Shen et al. (2023). In addition to GUIDE, we evaluate two additional variants to better ablate the effects of each stage of GUIDE. EMM + cal refers to EMM that is trained on the salient layers found in GUIDE's uncertainty calibration stage instead of arbitrarily selected layers. EMM + curric refers to EMM that is trained upon the curriculum data generated by GUIDE using the standard EDL loss Shen et al. (2023). Experimental details, training procedures, hyperparameter choices, adversarial attack configurations, and dataset information are provided in Appendix C.

## 5.1 CORE RESULTS

For our core set of experiments, we evaluate the accuracy, ID coverage, OOD detection, including both near- and far-OOD settings, adversarial robustness, AUROC, and adversarial AUROC of both GUIDE and its variants, and comparative baselines across a diverse suite of ID/OOD dataset pairs, in order to assess overall reliability under distributional shift and attack. ID/OOD/Adv coverage is computed using a binary rejection threshold based upon the AUROC, see Appendix C.2 for further details. These results are summarised in Table 1 and further analysis can be found in Appendix B.1.

---

[2] Our open-source repository is available at: https://anonymous.4open.science/r/guide-C9E7/README.md

[3] Details of the maximum L2PGD perturbation for each dataset are discussed in Appendix C.6

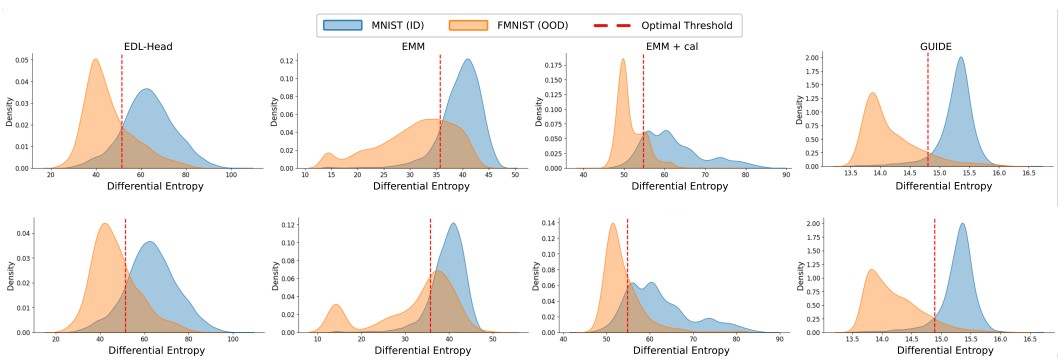

Figure 2: Visualised AUROC plots (top row) and Adv AUROC plots (bottom row), including the binary decision threshold for OOD/Adv rejection, for comparative methods where the ID dataset is MNIST and the OOD dataset is FashionMNIST.

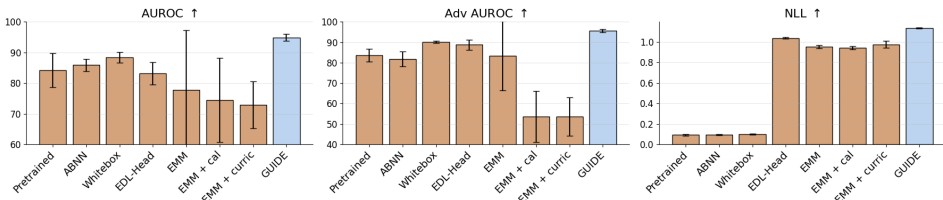

Figure 3: AUROC, adversarial AUROC, and negative log likelihood across comparative approaches where the ID → OOD dataset is MNIST and FashionMNIST. High AUROC and NLL is desirable, to show better ID → OOD/Adv separation and reduced overconfidence. GUIDE is in blue.

Firstly, across all datasets, every approach (including GUIDE and its variants) maintains near-ceiling accuracy across all datasets (e.g., 90-99%), showing that none of the UQ approaches compromise classification performance on clean ID data downstream from the pretrained model. This finding concretely shows that any observed gains in robustness through GUIDE are not due to degradation in performance.

In terms of ID coverage (the proportion of ID inputs retained after abstention), the pretrained base model rarely exceeds $80\%$ (e.g., $79.26\%$ on MNIST → FashionMNIST). Intrusive methods such as ABNN raise coverage (up to $85.91\%$) but retain high OOD ($\geq 20\%$) and adversarial inputs. Post-hoc baselines like EMM push ID coverage above $90\%$, yet still admit excessive adversarial examples ($\geq 20\%$). In contrast, GUIDE achieves a better balance: strong ID coverage ($\sim 87\%$) while cutting OOD to $\leq 8\%$ and adversarial to $\leq 5\%$, delivering reliable abstention without sacrificing retention.

Intrusive methods such as ABNN often fail to reject harmful inputs, with OOD coverage exceeding $20\%$ and adversarial coverage surpassing $40\%$ across multiple benchmarks (e.g., $46.27\%$ adversarial coverage on MNIST → KMNIST). Post-hoc baselines like EMM show some improvement, but still retain a substantial fraction of adversarial inputs ($\geq 20\%$). In contrast, GUIDE consistently reduces OOD coverage to below $10\%$ (e.g., $6.78\%$ on MNIST → KMNIST) and adversarial coverage to below $5\%$ (e.g., $4.60\%$ on MNIST → FashionMNIST), highlighting its ability to reliably abstain on uncertain or adversarial inputs while preserving ID performance.

In terms of AUROC (measure of ID/OOD separability), post-hoc baselines such as Whitebox and EMM achieve moderate separation between ID and OOD inputs (e.g., $\approx 77\%$ AUROC on MNIST → FashionMNIST), but their adversarial AUROC remains inconsistent ($\leq 85\%$). Intrusive methods like ABNN perform similarly, rarely exceeding $86\%$ AUROC. In contrast, GUIDE consistently delivers near-perfect discrimination, achieving $94.85\%$ AUROC and $95.72\%$ adversarial AUROC on MNIST → FashionMNIST, and above $95\%$ on MNIST → KMNIST. These results highlight GUIDE's ability to sharply distinguish in-distribution from both OOD and adversarial inputs, substantially outperforming prior intrusive and post-hoc methods.

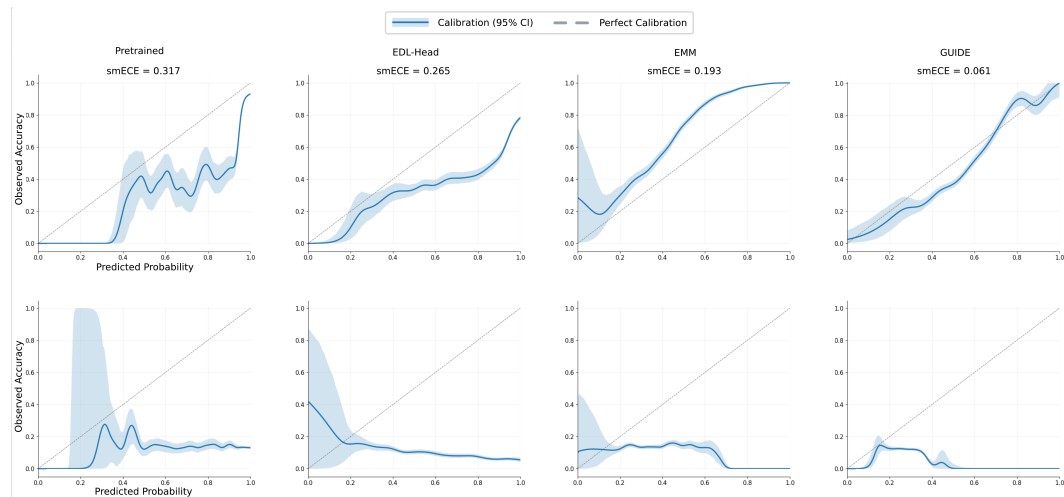

Figure 4: Calibration plots and smoothed expected calibration error (smECE) for ID (MNIST, top) and OOD (FashionMNIST, bottom) for comparative methods. The pretrained model is overconfident, EMM is underconfident, while GUIDE shows good calibration.

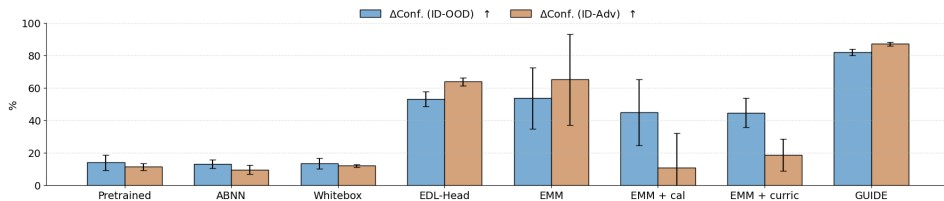

Figure 5: Mean drop in confidence between ID → OOD data and ID → Adv data where the the ID → OOD dataset is MNIST and FashionMNIST. GUIDE shows the largest drop in confidence between ID and OOD/Adv data validating how it learns uncertainty across distributional shifts.

We further ablate GUIDE with two variants: EMM + cal, which uses GUIDE's saliency-based calibration for layer selection, and EMM + curric, which applies the noise-driven curriculum without soft targets or custom loss. EMM + cal yields modest gains over EMM, showing that informed feature selection helps somewhat. EMM + curric often degrades performance, as corrupted views without guidance induce over- or underconfidence. GUIDE, combining all components, consistently outperforms both, demonstrating that all components are essential.

As shown in Figure 3, existing methods vary across AUROC, adversarial AUROC, and NLL (higher NLL indicates less overconfidence). Intrusive models like ABNN improve over baseline but plateau below 90%, while post-hoc methods such as Whitebox achieve competitive AUROC yet mainly reduce overconfidence, reflected in higher NLL. GUIDE reaches near-95% AUROC and adversarial AUROC with the highest NLL, offering reliable uncertainty. Figure 2 confirms this, as GUIDE shows the clearest ID/OOD separation, unlike the overlapping curves of EDL-Head and EMM.

Figure 4 shows calibration curves highlighting the limits of existing methods. The pretrained model and EDL-Head fall below the diagonal on ID data, indicating overconfidence (smECE = 0.317 and 0.265), and assign high probabilities to OOD inputs. EMM, by contrast, lies above the diagonal, underconfident on ID data (smECE = 0.193) yet still overly confident on OOD inputs. GUIDE instead aligns closely with the diagonal (smECE = 0.061) while keeping OOD probabilities low, demonstrating reliable calibration and effective OOD rejection.

Figure 5 shows the drop in predictive confidence between ID and shifted inputs. The pretrained model, ABNN, and Whitebox exhibit almost no drop ($\leq 15\%$), remaining overconfident on OOD and adversarial data. EDL-Head and EMM achieve larger drops ($50 \rightarrow 65\%$) but are unstable. In

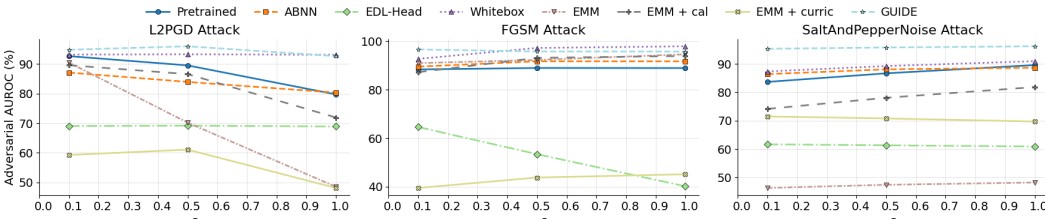

Figure 6: Adversarial AUROC (%) across varying perturbation strengths ($\epsilon$) for three attack types (L2PGD, FGSM, and Salt and Pepper noise). Higher AUROC indicates better robustness.

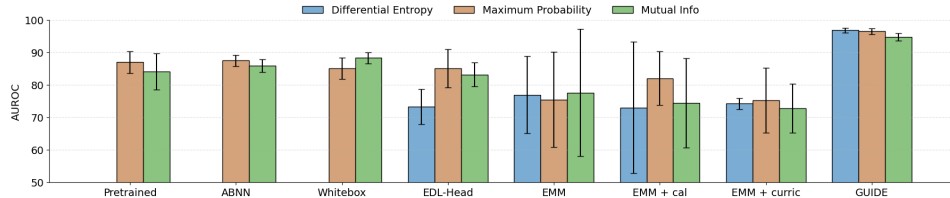

Figure 7: AUROC for different ID-OOD threshold metrics (differential entropy, maximum probability, mutual information) where the ID dataset is MNIST, and the OOD dataset is FashionMNIST.

contrast, GUIDE yields the largest and most consistent drops ($\approx 82\%$ ID/OOD, $\approx 87\%$ ID/Adv), demonstrating its ability to retain confidence on ID samples while reducing it on harmful inputs.

Our evaluation highlights a key distinction between intrusive and post-hoc methods. Intrusive approaches like ABNN and EDL-Head can achieve strong AUROC but require modifying or retraining the base model, which is often impractical in large-scale or black-box settings. Post-hoc methods are more general since they operate on frozen networks but tend to be unstable. GUIDE bridges this gap, retaining post-hoc accessibility while matching or surpassing intrusive baselines in robustness.

## 5.2 ADVERSARIAL ATTACK ANALYSIS

Figure 6 shows adversarial AUROC across perturbation strengths $\epsilon$ for L2PGD, FGSM, and Salt-and-Pepper attacks. GUIDE sustains high AUROC ($\geq 90\%$) across all perturbations, while pretrained, ABNN, and EMM models often drop below 70%. Robustness holds even under non-gradient Salt-and-Pepper noise, highlighting the effectiveness of GUIDE's saliency-calibrated curriculum in preserving adversarial separability across both gradient-based and gradient-free settings.

## 5.3 THRESHOLD ANALYSIS

To assess robustness across uncertainty metrics, we compare AUROC under differential entropy, maximum probability, and mutual information (Figure 7). Pretrained, ABNN, and Whitebox models show moderate but inconsistent performance, while EDL-Head and EMM are even less stable, often dropping below $80\%$ with wide error bars. In contrast, GUIDE maintains consistently high AUROC ($\geq 95\%$) with minimal variance, demonstrating reliable uncertainty estimates.

## 6 CONCLUSION AND FUTURE WORK

GUIDE is a post-hoc evidential meta-model that attaches to any pretrained classifier to explicitly learn when and how much to be uncertain. Through saliency-driven calibration, a noise-based curriculum, and an uncertainty-aware loss, GUIDE achieves reliable ID→OOD/Adversarial separation without retraining or altering the base model. Extensive experiments across diverse datasets and adversarial settings show that GUIDE consistently outperforms both intrusive and post-hoc baselines in AUROC, coverage, and calibration. Despite its simplicity, GUIDE remains lightweight, architecture-agnostic, and practical for real-world deployment. Future work includes extending GUIDE to regression and structured prediction tasks, exploring its integration with active learning and safe decision-making.

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

## A THEORETICAL ANALYSIS

This appendix provides the proofs to complement Theorem 1 presented in Section 4. More specifically, we provide a proof for how much information is retained from the pretrained model during GUIDE's saliency-based calibrations stage.

### A.1 THEOREM 1

**Theorem 1.** Let $\eta \in (0, 1]$ denote the saliency coverage threshold, $\kappa \in (0, 1]$ the alignment of saliency scores with the Fisher information mass, and $\rho \in (0, 1]$ the information preserved by the projection. Under local linearisation and restricting to cross-depth correlated blocks, GUIDE's saliency calibration retains at least $\rho\kappa\eta$ of the pretrained Fisher trace with respect to $\beta$. Moreover, if the saliency normalisation is stable and the projections are well-conditioned, then $\kappa \approx 1$ and $\rho \approx 1$, so that the retained Fisher fraction is essentially determined by $\eta$.

*Proof.* Let $f_\theta : \mathcal{X} \to \mathbb{R}^K$ be a pretrained network with frozen parameters $\theta$. On the train dataset $\mathcal{D}$, work in the standard local linear regime:

$$z(x) = b + \sum_{\ell=1}^{L} W_\ell \Phi_\ell(x) \quad \text{in } L^2(\mathcal{D}) \tag{8}$$

where $\Phi_\ell(x)$ are layer features (larger $\ell$ is closer to the networks output). Two layers are said to be cross-depth correlated if their feature covariance is non-zero. Form the undirected graph whose vertices are layers and whose edges connect pairs $(i, j)$ with $\mathrm{Cov}(\Phi_i, \Phi_j) \neq 0$. Let $\mathcal{I}_{\mathrm{corr}}$ be the union of connected components that contain at least one edge spanning different depths. All "sum over all layers" denominators below sum only over $\mathcal{I}_{\mathrm{corr}}$, so purely uncorrelated blocks are ignored by design.

GUIDE computes non-negative saliencies $M_\ell$, sorts layers by $M_\ell$ in descending order, and selects the smallest prefix $L_{\mathrm{sal}}$ that satisfies the coverage constraint in Equation 3. For each selected layer $\ell \in \mathcal{S} := L_{\mathrm{sal}} \cap \mathcal{I}_{\mathrm{corr}}$ in the correlated set, the branch $P_\ell : \mathbb{R}^{d_\ell} \to \mathbb{R}^K$ produces $\psi_\ell = P_\ell \Phi_\ell$, and $\Psi$ concatenates $\{\psi_\ell\}_{\ell \in \mathcal{S}}$. For analysis, we replace $\Phi_\ell$ by its innovation $\tilde{\Phi}_\ell$ and study $\psi_\ell = P_\ell \tilde{\Phi}_\ell$; this removes components that are linearly predictable from lower-depth information and therefore can only decrease Fisher, yielding a valid lower bound. [4]

Next is to ensure that the features from different correlated layers contribute non-redundant information to the Fisher calculation. To do this, we construct innovations by processing the correlated layers in order of depth. We work in the Hilbert space $L^2(\Omega, \mathbb{P})$ with inner product $\langle A, B \rangle = \mathbb{E}[A^\top B]$. Let the indices in $\mathcal{I}_{\mathrm{corr}}$ be ordered from lower to higher depth, $i_1 > i_2 > \ldots > i_m$ (so $i_1$ is closest to the output). At each step $t$, we remove from $\Phi_{i_t}$ the part that can be explained as a linear combination of the innovations from all later depths (i.e. those processed earlier in the ordering). Formally, if $\prod_{t-1}$ denotes the orthogonal projector in $L^2$ onto $\mathrm{span}\{\Phi_{i_s} : s < t\}$, then the innovation at depth $i_t$ is defined by $\tilde{\Phi}_{i_t} := \Phi_{i_t} - \prod_{t-1} \Phi_{i_t}$. By construction, innovations from different depths are orthogonal in $L^2$, meaning $\mathbb{E}[\tilde{\Phi}_{i_s} \tilde{\Phi}_{i_t}^\top] = 0$ whenever $s \neq t$. We denote by $\sum_{i_t}^{\mathrm{inn}} = \mathrm{Cov}(\tilde{\Phi}_{i_t}) \succeq 0$ the covariance of the innovation at depth $i_t$. This orthogonalisation isolates the new, linearly independent contribution of each correlated layer after accounting for deeper ones. It makes the Fisher matrix block-diagonal, so the total trace is just the sum over innovations; without it, cross-layer covariances would produce off-diagonal terms and break this additivity.

We model the GUIDE setting as a local Gaussian parametric experiment, in which the output vector $U$ depends linearly on the depth-ordered innovations, with additive Gaussian noise of fixed covariance:

$$U \mid \tilde{\Phi} \sim \mathcal{N}\big(b + \sum_{t=1}^{m} B_{i_t} \tilde{\Phi}_{i_t}, \Sigma_\epsilon\big) \qquad \Sigma_\epsilon \succeq 0 \tag{9}$$

---

[4]Each branch $g_\ell : \mathbb{R}^{d_\ell} \to \mathbb{R}^K$ is linear in our method; we therefore write $g_\ell(\Phi_\ell) = P_\ell \Phi_\ell$, $P_\ell \in \mathbb{R}^{K \times d_\ell}$. If $g_\ell$ includes non-linearities, all Fisher bounds below apply to its local linearisation, with $P_\ell$ denoting the corresponding linear operator.

The unknown parameter $\beta := \{B_{i_t}\}_{t=1}^m$ i.e., the collection of block matrices mapping each innovation $\tilde{\Phi}_{i_t}$ to the output. This model makes the classical Fisher information exactly computable from the innovation covariances $\sum_{i_t}^{\text{inn}}$, and constant factors from $\Sigma_\epsilon$ cancel in all Fisher ratios of interest. Because the innovations are uncorrelated across $t$, the Fisher matrix is block diagonal $\mathcal{I}(\beta) = \text{blkdiag}_t(\Sigma_\epsilon^{-1} \otimes \sum_{i_t}^{\text{inn}})$ whose trace gives the total Fisher from all correlated blocks:

$$\mathfrak{F}_{\text{all}} = \text{tr}\,\mathcal{I}(\beta) = \text{tr}(\Sigma_\epsilon^{-1}) \sum_{\ell \in \mathcal{I}_{\text{corr}}} \text{tr}(\Sigma_\ell^{\text{inn}}) \tag{10}$$

The meta-model only receives features from the selected layers $\mathcal{S} = L_{\text{sal}} \cap \mathcal{I}_{\text{corr}}$. For each such layer $\ell$, the innovation $\tilde{\Phi}_\ell$ is first projected through the learned branch $P_\ell$ to give $\psi_\ell = P_\ell \tilde{\Phi}_\ell$ for $\ell \in \mathcal{S}$. Parametrising:

$$U \mid \psi \sim \mathcal{N}\big(b + \sum_{\ell \in \mathcal{S}} B_{i_t} C_\ell \psi_\ell, \Sigma_\epsilon\big) \qquad \Sigma_\epsilon \succeq 0 \tag{11}$$

with $\{C_\ell\}_{\ell \in \mathcal{S}}$ in place of $\{B_\ell\}$- one matrix per projected branch- gives:

$$\mathcal{I}_\Psi(C) = \text{blkdiag}_{\ell \in \mathcal{S}}(\Sigma_\epsilon^{-1} \otimes P_\ell \Sigma_\ell^{\text{inn}} P_\ell^\top), \qquad \mathfrak{F}_\Psi = \text{tr}\,\mathcal{I}_\Psi(C) = \text{tr}(\Sigma_\epsilon^{-1}) \sum_{\ell \in \mathcal{S}} \text{tr}(P_\ell \Sigma_\ell^{\text{inn}} P_\ell^\top) \tag{12}$$

where $\mathfrak{F}_\Psi$ is the total Fisher content accessible to the meta-model. It is important to measure, in worst-case relative terms, how much of the per-layer Fisher content is preserved by the projection. We define the projection–retention factor:

$$\rho := \inf_{\ell \in \mathcal{S}} \frac{\text{tr}(P_\ell \Sigma_\ell^{\text{inn}} P_\ell^\top)}{\text{tr}(\Sigma_\ell^{\text{inn}})} \quad \in (0, 1] \tag{13}$$

which quantifies the minimum per-branch retention of Fisher information after projection. By construction, $\rho$ is the smallest fraction of Fisher (in trace form) that any selected branch retains compared to the unprojected innovation. From this definition, the total projected Fisher content can be bounded as:

$$\sum_{\ell \in \mathcal{S}} \text{tr}(P_\ell \Sigma_\ell^{\text{inn}} P_\ell^\top) \geq \rho \sum_{\ell \in \mathcal{S}} \text{tr}(\Sigma_\ell^{\text{inn}})) \tag{14}$$

This inequality states that, in aggregate, the selected meta-model branches retain at least a fraction $\rho$ of the Fisher content they would have had without projection.

For the purposes of the bound, the *true* per-layer contribution to the Fisher trace is measured by $M_\ell^* = \text{tr}(\Sigma_\ell^{\text{inn}})$, which we refer to as the ideal Fisher weight of layer $\ell$. In practice, GUIDE does not have access to $M_\ell^*$ and instead uses computed emperical saliency scores $M_\ell$ to select the subset $L_{\text{sal}}$ via the coverage rule described earlier. To connect the empirical selection to the ideal Fisher weighting, we assume a calibrated coverage condition:

$$\sum_{\ell \in \mathcal{S}} M_\ell^* \geq \kappa\,\eta \sum_{\ell \in \mathcal{I}_{\text{corr}}} M_\ell^* \qquad \kappa \in (0, 1] \tag{15}$$

where $\eta$ is the coverage threshold from the selection rule, and $\kappa$ quantifies any mismatch between the empirical saliencies and the ideal Fisher weights. When the base network is exactly linear (or locally linear with fixed gates and controlled rescalings), the empirical saliencies are proportional to $M_\ell^*$ and $\kappa = 1$. In more general settings, $\kappa$ may be smaller; in that case, it can be treated as a one-off model-dependent calibration factor, absorbed into the effective coverage level $\eta$.

Finally, since both $\mathfrak{F}_\Psi$ and $\mathfrak{F}_{\text{all}}$ include the factor $\text{tr}(\Sigma_\epsilon^{-1})$, it cancels in the ratio. By the definition of $\rho$,

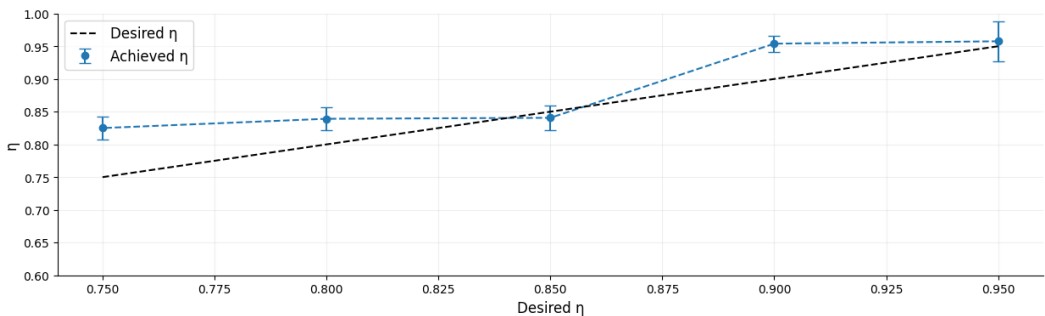

Figure 8: Visual approximation of Theorem 1: achieved cumulative relevance coverage ($\eta$) under saliency calibration versus the desired target. The empirical estimates, obtained via Jacobian-based Fisher information, closely track or exceed the theoretical bound.

$$\frac{\sum_{\ell \in \mathcal{S}} \text{tr}(P_\ell \Sigma_\ell^{\text{inn}} P_\ell^\top)}{\sum_{\ell \in \mathcal{I}_{\text{corr}}} \text{tr}(\Sigma_\ell^{\text{inn}})} \geq \rho \cdot \frac{\sum_{\ell \in \mathcal{S}} \text{tr}(\Sigma_\ell^{\text{inn}})}{\sum_{\ell \in \mathcal{I}_{\text{corr}}} \text{tr}(\Sigma_\ell^{\text{inn}})} \tag{16}$$

The calibrated coverage condition ensures the second fraction is at least $\kappa\eta$, giving:

$$\frac{\mathfrak{F}_\Psi}{\mathfrak{F}_{\text{all}}} \geq \rho\kappa\eta \tag{17}$$

and hence $\mathfrak{F}_\Psi \geq (\rho\kappa\eta)\mathfrak{F}_{\text{all}}$.

This shows that, even after restricting to the $\eta$-coverage saliency set and applying potentially lossy $K$-dimensional projections, the GUIDE head retains at least a fraction $\rho\kappa\eta$ of the total classical Fisher information available from all correlated blocks. In other words, the selected layers preserve a guaranteed share of the information about the linearised parameter $\beta$.

$\square$

## B  ADDITIONAL EXPERIMENTS

This appendix provides additional experimental insights and analyses to complement the core results presented in Section 5. Specifically, we include extended analysis of metrics, adversarial attacks, thresholds, and ablations of hyperparameters introduced in GUIDE.

### B.1  EXTENDED ANALYSIS OF CORE RESULTS

To empirically assess Theorem 1, we estimate the achieved cumulative relevance coverage ($\eta$) using Jacobian-based Fisher information on the MNIST $\rightarrow$ FashionMNIST. Figure 8 compares the achieved values to the desired targets. For target values of $\eta \in \{0.75, 0.80, 0.85\}$, the calibration procedure consistently selected {pool1 and conv1} as the intermediate layers, yielding achieved $\eta$ slightly above the desired level. At higher targets ($\eta = 0.90, 0.95$), the set expanded to include pool2, which further increased coverage and again exceeded the bound predicted by Theorem 1. These results confirm that the saliency calibration reliably preserves the intended fraction of Fisher information and, if anything, errs on the side of retaining more information than required, thereby validating the practical soundness of our theoretical guarantee.

A clear distinction between EMM and GUIDE emerges when comparing the intermediate features each method relies on (Figure 9, Table 2). In the evidential meta-model, EMM depends on arbitrarily chosen layers (shown in yellow), which may not align with the most relevant or informative features. Moreover, selecting layers manually becomes increasingly impractical as models scale in depth and complexity, making an automated calibration procedure essential for general applicability. By contrast, GUIDE calibrates layer selection through cumulative relevance coverage, consistently

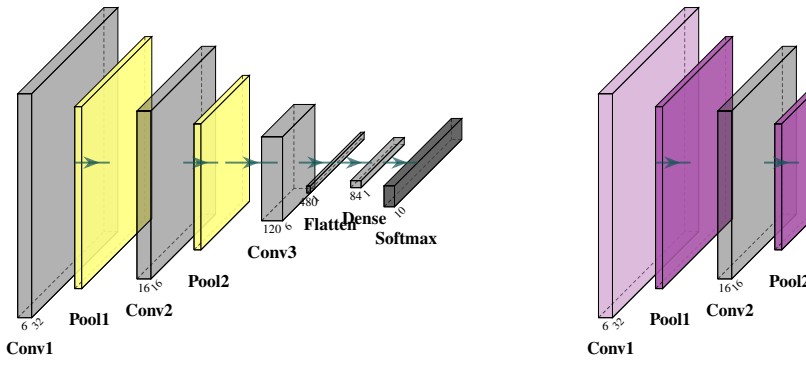

(a) Evidential Meta-Model      (b) GUIDE meta-model

Figure 9: Qualitative comparison between the intermediate features selected in the LeNet model for (a) Evidential meta-model and (b) GUIDE meta-model. In the **Evidential meta-model**, intermediate layers are selected arbitrarily, shown in **yellow**, whereas in the **GUIDE meta-model**, the chosen features are calibrated to correspond to the most relevant and salient layers, highlighted in **purple**.

Table 2: Comparison of layers selected by EMM and layers calibrated by GUIDE ($\eta = 0.9$, in order of relevancy) across different base networks. GUIDE consistently selects salient layers identified by relevance coverage, whereas EMM relies on arbitrary intermediate-layer choices.

| Base Model | EMM Selected Layers | GUIDE Calibrated Layers |
|---|---|---|
| LeNet (MNIST) | {pool1, pool2} | {pool1, conv1, pool2} |
| ResNet-18 (CIFAR-10) | {block1_3_out, block2_3_out, block3_3_out} | {block1_3_out, block2_3_out, block3_3_out} |
| SENet (CIFAR-100, Flowers) | {pool1, pool2, gap } | {gap, pool2} |

identifying salient layers (highlighted in **purple**). This behaviour is evident across architectures: whereas EMM typically relies on handpicked layers, GUIDE selects a compact set of semantically meaningful layers (e.g., `pool1`, `pool2`, and `conv1` in LeNet), ensuring the meta-model operates on features with maximal relevance. This principled calibration removes the arbitrariness of EMM's design and underpins GUIDE's improved stability and robustness across datasets.

The extended results for AUROC, adversarial AUROC, and NLL across multiple dataset pairs are reported in Figure 10 and Table 3. Consistent with the main results, intrusive baselines such as ABNN and EDL-Head achieve modest gains in AUROC but generally fail to reduce overconfidence across all settings. Post-hoc baselines like Whitebox and EMM often attain higher NLL and competitive

Table 3: The mean NLL, drop in confidence between ID → OOD data, and ID → Adv data with 95% CI of the comparative approaches with a variety of ID and OOD datasets. Highlighted cells denote the best performance for each metric. * indicates datasets classed as Near-OOD.

| | Pretrained | ABNN | EDL-Head | Whitebox | EMM | EMM + cal | EMM + curric | GUIDE |
|---|---|---|---|---|---|---|---|---|
| **MNIST → FashionMNIST** | | | | | | | | |
| NLL ↑ | 0.09 [0.09, 0.10] | 0.09 [0.09, 0.10] | 1.04 [1.03, 1.04] | 0.10 [0.09, 0.10] | 0.95 [0.94, 0.96] | 0.94 [0.93, 0.95] | 0.97 [0.95, 1.00] | 1.13 [1.12, 1.14] |
| ΔConf. (ID-OOD) ↑ | 0.14 [0.10, 0.18] | 0.13 [0.11, 0.15] | 0.53 [0.50, 0.57] | 0.14 [0.09, 0.18] | 0.54 [0.40, 0.68] | 0.45 [0.30, 0.61] | 0.45 [0.38, 0.52] | 0.82 [0.79, 0.85] |
| ΔConf. (ID-Adv) ↑ | 0.11 [0.10, 0.13] | 0.10 [0.08, 0.12] | 0.64 [0.62, 0.66] | 0.12 [0.11, 0.13] | 0.65 [0.44, 0.86] | 0.11 [−0.05, 0.27] | 0.19 [0.11, 0.26] | 0.87 [0.86, 0.89] |
| **MNIST → KMNIST** | | | | | | | | |
| ID NLL ↑ | 0.09 [0.08, 0.09] | 0.09 [0.09, 0.10] | 1.04 [1.03, 1.05] | 0.09 [0.08, 0.10] | 0.95 [0.94, 0.96] | 0.96 [0.94, 0.97] | 0.97 [0.94, 0.99] | 1.13 [1.12, 1.13] |
| ΔConf. (ID-OOD) ↑ | 0.17 [0.16, 0.17] | 0.18 [0.17, 0.19] | 0.62 [0.61, 0.64] | 0.17 [0.17, 0.18] | 0.69 [0.63, 0.75] | 0.75 [0.70, 0.80] | 0.61 [0.59, 0.63] | 0.82 [0.80, 0.84] |
| ΔConf. (ID-Adv) ↑ | 0.08 [0.07, 0.08] | 0.06 [0.06, 0.07] | 0.63 [0.61, 0.65] | 0.10 [0.09, 0.11] | 0.17 [−0.15, 0.50] | 0.15 [−0.08, 0.38] | 0.15 [0.10, 0.21] | 0.81 [0.79, 0.84] |
| **MNIST → EMNIST\*** | | | | | | | | |
| ID NLL ↑ | 0.09 [0.08, 0.09] | 0.10 [0.09, 0.10] | 1.04 [1.03, 1.04] | 0.10 [0.09, 0.11] | 0.95 [0.93, 0.97] | 0.95 [0.94, 0.95] | 0.97 [0.96, 0.98] | 1.13 [1.12, 1.13] |
| ΔConf. (ID-OOD) ↑ | 0.16 [0.15, 0.17] | 0.17 [0.15, 0.18] | 0.59 [0.58, 0.60] | 0.16 [0.15, 0.17] | 0.49 [0.45, 0.53] | 0.58 [0.54, 0.62] | 0.47 [0.43, 0.51] | 0.67 [0.65, 0.69] |
| ΔConf. (ID-Adv) ↑ | 0.10 [0.09, 0.10] | 0.08 [0.07, 0.08] | 0.63 [0.62, 0.65] | 0.11 [0.10, 0.12] | 0.20 [−0.03, 0.43] | 0.28 [0.13, 0.42] | 0.19 [0.13, 0.24] | 0.75 [0.73, 0.77] |
| **CIFAR10 → SVHN** | | | | | | | | |
| ID NLL ↑ | 1.83 [1.61, 2.05] | 1.63 [1.63, 1.63] | 1.16 [1.15, 1.18] | 1.80 [1.58, 2.03] | 1.00 [0.75, 1.24] | 0.65 [0.63, 0.68] | 0.62 [0.60, 0.65] | 1.17 [1.14, 1.20] |
| ΔConf. (ID-OOD) ↑ | 0.09 [0.05, 0.14] | −0.03 [−0.04, −0.03] | 0.60 [0.50, 0.70] | 0.12 [0.09, 0.15] | 0.52 [0.33, 0.71] | 0.33 [0.21, 0.45] | 0.41 [0.29, 0.54] | 0.72 [0.59, 0.86] |
| ΔConf. (ID-Adv) ↑ | −0.02 [−0.03, −0.02] | −0.03 [−0.04, −0.03] | 0.58 [0.47, 0.70] | −0.03 [−0.04, −0.01] | 0.37 [0.32, 0.43] | −0.01 [−0.08, 0.07] | 0.07 [−0.06, 0.20] | 0.48 [0.31, 0.64] |
| **CIFAR10 → CIFAR100\*** | | | | | | | | |
| ID NLL ↑ | 2.41 [2.25, 2.56] | 1.69 [1.69, 1.70] | 1.30 [1.28, 1.32] | 1.12 [0.96, 1.28] | 1.01 [0.80, 1.22] | 0.92 [0.89, 0.95] | 1.30 [0.67, 1.92] | 1.34 [1.33, 1.35] |
| ΔConf. (ID-OOD) ↑ | 0.09 [0.08, 0.09] | 0.10 [0.10, 0.11] | 0.49 [0.48, 0.49] | 0.23 [0.18, 0.28] | 0.28 [0.21, 0.34] | 0.36 [0.33, 0.39] | 0.11 [−0.42, 0.64] | 0.55 [0.53, 0.57] |
| ΔConf. (ID-Adv) ↑ | 0.03 [0.02, 0.03] | 0.00 [−0.01, 0.00] | 0.58 [0.56, 0.61] | 0.34 [0.30, 0.39] | 0.24 [0.14, 0.35] | 0.33 [0.29, 0.37] | 0.26 [0.10, 0.41] | 0.64 [0.62, 0.65] |
| **Oxford Flowers → Deep Weeds** | | | | | | | | |
| ID NLL ↑ | 1.31 [1.10, 1.52] | 3.75 [3.74, 3.76] | 3.00 [2.99, 3.01] | 1.40 [0.92, 1.87] | 3.87 [3.60, 4.15] | 3.55 [3.26, 3.85] | 3.91 [3.90, 3.92] | 3.29 [3.28, 3.30] |
| ΔConf. (ID-OOD) ↑ | 0.03 [0.00, 0.05] | 0.13 [0.03, 0.23] | 0.33 [−0.09, 0.76] | 0.03 [−0.07, 0.12] | 0.10 [0.04, 0.16] | 0.13 [0.06, 0.20] | 0.18 [0.08, 0.29] | 0.10 [−0.17, 0.38] |
| ΔConf. (ID-Adv) ↑ | 0.04 [0.01, 0.06] | 0.14 [0.06, 0.22] | 0.43 [0.06, 0.79] | 0.07 [−0.04, 0.19] | 0.12 [0.07, 0.18] | 0.17 [0.10, 0.24] | 0.22 [0.12, 0.31] | 0.19 [−0.12, 0.50] |

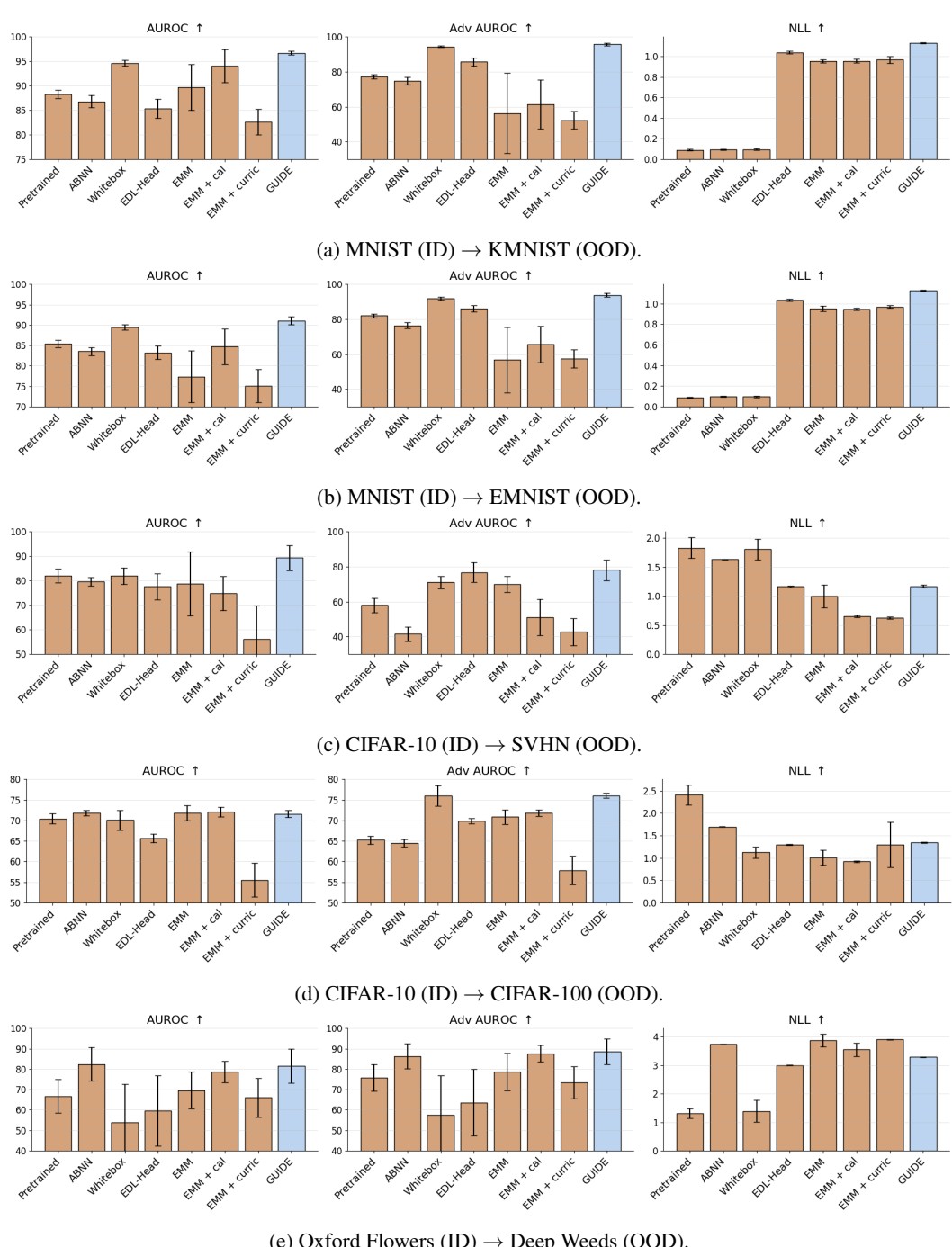

(a) MNIST (ID) → KMNIST (OOD).

(b) MNIST (ID) → EMNIST (OOD).

(c) CIFAR-10 (ID) → SVHN (OOD).

(d) CIFAR-10 (ID) → CIFAR-100 (OOD).

(e) Oxford Flowers (ID) → Deep Weeds (OOD).

Figure 10: Mean drop in confidence between ID → OOD data and ID → Adv data for each evaluated dataset pair.

AUROC, yet remain unstable: EMM in particular shows large variance across runs, with AUROC values sometimes below 80% and adversarial AUROC highly inconsistent. In contrast, GUIDE achieves the highest AUROC on both near-OOD (e.g., MNIST → KMNIST, CIFAR-10 → CIFAR-100) and far-OOD settings (e.g., CIFAR-10 → SVHN, Oxford Flowers → DeepWeeds), while consistently producing the high NLL, indicating well-calibrated confidence. Importantly, GUIDE also delivers the largest drop in confidence between ID and both OOD and adversarial inputs across all dataset pairs, confirming its ability to reject harmful inputs while preserving in-distribution

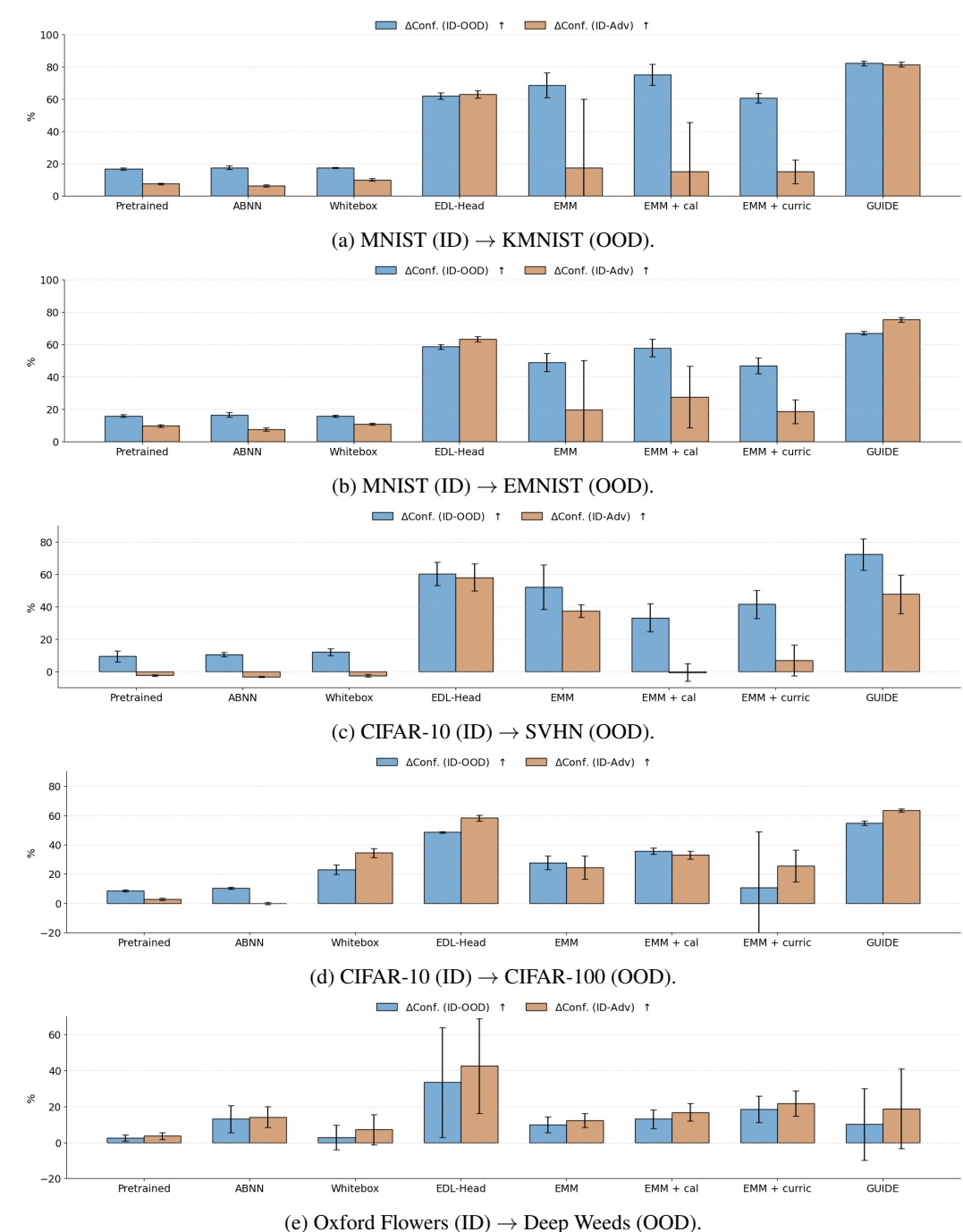

(a) MNIST (ID) → KMNIST (OOD).

(b) MNIST (ID) → EMNIST (OOD).

(c) CIFAR-10 (ID) → SVHN (OOD).

(d) CIFAR-10 (ID) → CIFAR-100 (OOD).

(e) Oxford Flowers (ID) → Deep Weeds (OOD).

Figure 11: Mean drop in confidence between ID → OOD data and ID → Adv data for all evaluated upon dataset pairs.

predictions. These trends hold across both simple (LeNet, MNIST) and deeper architectures (ResNet-18, DenseNet), demonstrating that GUIDE scales effectively and remains robust under diverse distribution shifts.

The confidence drop analysis in Figure 11 provides further evidence of GUIDE's effectiveness across all evaluated dataset pairs. For MNIST → KMNIST and MNIST → EMNIST (Figures 11a and 11b), the pretrained model, ABNN, and Whitebox exhibit almost no drop in confidence ($\leq 15\%$), confirming their tendency to remain overconfident under shift. EDL-Head and EMM achieve larger

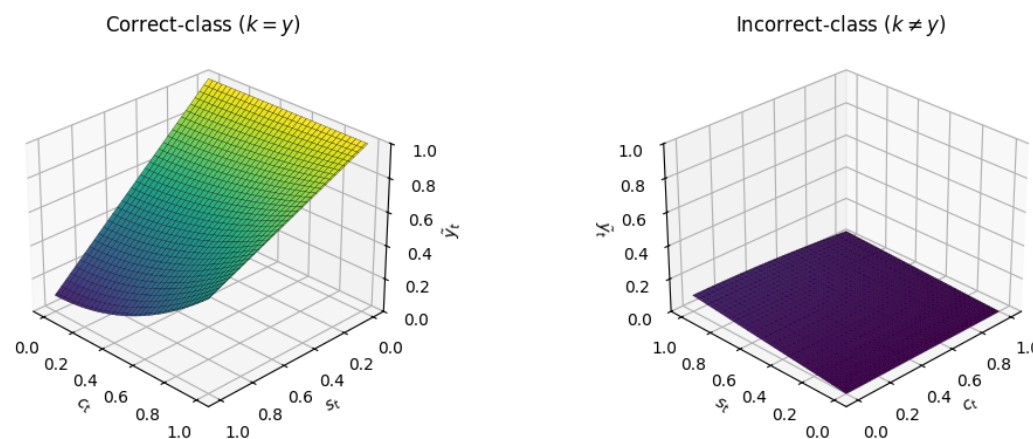

Figure 12: Visualization of the soft-target assignment $\tilde{y}_k$ under our weighting scheme as a function of model confidence $c_t$ and noise $s_t$.

drops ($40 \rightarrow 60\%$), but in some cases adversarial drops remain significantly smaller than OOD drops, indicating incomplete robustness. GUIDE clearly outperforms all baselines across the majority of dataset pairs, producing stable confidence drops above $80\%$ in some cases for both OOD and adversarial inputs, whereas competing methods rarely exceed $60\%$. Taken together, these results confirm that GUIDE systematically learns to maintain high confidence on ID samples while sharply reducing confidence on both OOD and adversarial data, a behaviour that neither intrusive nor post-hoc baselines reliably achieve.

Figure 12 illustrates how the GUIDE meta-model transforms the base model's confidence $c_t$ and the estimated noise level $s_t$ into adjusted training targets $\tilde{y}_t$ for both the correct and incorrect classes. Across the entire domain, $\tilde{y}_t \in [0, 1]$, ensuring that the constructed target vector is a valid probability distribution compatible with the cross-entropy objective. For the correct class ($k = y$), $\tilde{y}_t$ increases smoothly with confidence while being attenuated by the noise level. In low-noise, high-confidence regions, the target approaches 1, reinforcing learning of the correct label. Conversely, when confidence is low or noise is high, the assigned probability decreases, allowing GUIDE to down-weight uncertain or corrupted inputs. Importantly, in the downstream evidential Dirichlet formulation, such cases still correspond to low total evidence, preserving uncertainty even when mean confidence is high.

For incorrect classes ($k \neq y$), the surface remains flat and close to zero, preventing spurious reinforcement of misclassified outputs. In extreme high-noise, high-confidence settings, the assignment approaches uniform mass $1/K$ due to the GUIDE meta-model's uniform-mixing term. While this may appear as an undesirable "bump," it is in fact deliberate: it enforces maximum uncertainty under severe corruption, maintaining probabilistic validity and discouraging overconfident misclassifications. Although this corner case rarely arises in practice, it is crucial for ensuring GUIDE's theoretical guarantees.

Taken together, these surfaces demonstrate that GUIDE adaptively balances reinforcement of reliable predictions with suppression of noisy or misleading signals, yielding soft targets that are both probabilistically sound and robust to distributional corruption.

## B.2 Adversarial Attack Analysis

Across all attack types and perturbation strengths evaluated in Table 4, GUIDE consistently achieves the highest adversarial AUROC, with margins that are both statistically significant and robust across the 95% confidence intervals. Under the L2PGD attack, GUIDE outperforms all baselines at every perturbation level, reaching 96.07% at $\epsilon = 0.1$ and retaining a remarkably high 94.67% even at $\epsilon = 1.0$. By contrast, competing methods such as EMM and its calibrated or curriculum variants exhibit substantial degradation in performance, often falling below 60% AUROC under stronger perturbations.

Table 4: The mean adversarial AUROC with 95% CI of the comparative approaches for a variety of adversarial attacks and maximum perturbations $\epsilon$ where the ID dataset is MNIST and the OOD dataset is FashionMNIST. Highlighted cells denote the best performance for each metric.

| $\epsilon$ | Pretrained | ABNN | EDL-Head | Whitebox | EMM | EMM + cal | EMM + curric | GUIDE |
|---|---|---|---|---|---|---|---|---|
| | | | | L2PGD | | | | |
| 0.1 | 91.50 [89.95, 93.05]% | 89.54 [87.65, 91.43]% | 61.06 [46.05, 76.08]% | 90.70 [86.01, 95.38]% | 88.66 [83.87, 93.45]% | 80.70 [67.01, 94.39]% | 51.09 [36.44, 65.73]% | 96.07 [94.52, 97.61]% |
| 0.5 | 89.68 [88.43, 90.94]% | 87.18 [84.27, 90.09]% | 60.15 [44.48, 75.82]% | 90.93 [86.82, 95.04]% | 64.02 [54.91, 73.14]% | 76.04 [61.81, 90.27]% | 51.44 [41.88, 60.99]% | 96.72 [95.52, 97.93]% |
| 1.0 | 83.67 [81.38, 85.95]% | 81.81 [79.13, 84.48]% | 58.69 [42.29, 75.09]% | 90.95 [87.33, 94.57]% | 43.82 [35.42, 52.21]% | 57.88 [41.04, 74.72]% | 44.06 [39.74, 48.39]% | 94.67 [93.20, 96.13]% |
| | | | | FGSM | | | | |
| 0.1 | 87.27 [84.19, 90.35]% | 86.62 [83.50, 89.75]% | 61.04 [52.14, 69.95]% | 92.66 [90.09, 95.22]% | 79.45 [63.80, 95.10]% | 76.11 [49.96, 102.26]% | 47.81 [39.83, 55.79]% | 95.67 [94.53, 96.81]% |
| 0.5 | 88.40 [86.85, 89.94]% | 88.52 [85.74, 91.31]% | 46.48 [36.49, 56.47]% | 95.56 [93.09, 98.04]% | 91.72 [86.04, 97.40]% | 72.20 [29.94, 114.47]% | 49.13 [44.13, 54.12]% | 94.35 [92.62, 96.08]% |
| 1.0 | 88.23 [87.22, 89.24]% | 89.06 [86.68, 91.44]% | 34.72 [24.52, 44.93]% | 96.92 [95.48, 98.35]% | 95.36 [93.48, 97.24]% | 72.33 [29.26, 115.40]% | 48.61 [41.97, 55.24]% | 94.19 [91.83, 96.55]% |
| | | | | Salt & Pepper | | | | |
| 0.1 | 83.60 [79.67, 87.52]% | 86.27 [81.34, 91.21]% | 61.27 [56.24, 66.31]% | 90.42 [86.41, 94.42]% | 65.32 [47.09, 83.56]% | 76.00 [62.93, 89.08]% | 63.62 [54.65, 72.59]% | 96.05 [94.40, 97.70]% |
| 0.5 | 86.88 [83.15, 90.61]% | 89.16 [84.63, 93.68]% | 61.30 [56.28, 66.32]% | 91.66 [88.02, 95.31]% | 66.45 [48.96, 83.94]% | 78.27 [64.88, 91.67]% | 64.33 [56.06, 72.61]% | 96.34 [94.84, 97.83]% |
| 1.0 | 89.63 [86.22, 93.04]% | 91.00 [86.71, 95.30]% | 61.47 [56.55, 66.39]% | 92.75 [89.59, 95.91]% | 67.15 [50.35, 83.95]% | 80.46 [66.55, 94.37]% | 65.09 [57.40, 72.77]% | 96.60 [95.27, 97.93]% |

Table 5: The mean accuracy, OOD detection, and adversarial attack detection performance with 95% CI of the comparative approaches with a variety of uncertainty metrics. The adversarial attack is an L2PGD attack. Highlighted cells denote the best performance for each metric.

| | Pretrained | ABNN | EDL-Head | Whitebox | EMM | EMM + cal | EMM + curric | GUIDE |
|---|---|---|---|---|---|---|---|---|
| | | | | Mutual Information | | | | |
| ID Acc ↑ | 99.80 [99.75, 99.85]% | 99.78 [99.74, 99.82]% | 99.43 [99.31, 99.55]% | 99.92 [99.83, 100.00]% | 99.55 [99.49, 99.60]% | 99.69 [99.62, 99.75]% | 99.21 [99.04, 99.38]% | 99.87 [99.82, 99.92]% |
| ID Cov ↑ | 79.26 [76.31, 82.22]% | 74.88 [69.46, 80.29]% | 77.44 [75.24, 79.65]% | 76.38 [69.77, 83.00]% | 90.58 [87.85, 93.31]% | 92.32 [91.26, 93.38]% | 79.46 [72.51, 86.42]% | 87.05 [85.24, 88.85]% |
| OOD Cov ↓ | 24.89 [18.30, 31.47]% | 18.23 [12.40, 24.07]% | 22.51 [18.78, 26.24]% | 13.50 [10.48, 16.52]% | 31.87 [14.07, 49.68]% | 35.69 [23.92, 47.47]% | 40.18 [30.49, 49.87]% | 7.34 [3.54, 11.14]% |
| Adv Cov ↓ | 25.84 [20.52, 31.16]% | 24.35 [17.79, 30.92]% | 13.86 [11.91, 15.80]% | 9.04 [4.36, 13.72]% | 22.18 [6.91, 37.44]% | 55.47 [46.03, 64.91]% | 62.11 [51.81, 72.41]% | 4.60 [1.61, 7.58]% |
| AUROC ↑ | 84.18 [80.16, 88.21]% | 85.93 [84.50, 87.36]% | 83.23 [80.59, 85.86]% | 88.38 [86.22, 90.54]% | 77.68 [63.70, 91.66]% | 74.54 [64.69, 84.38]% | 72.85 [67.40, 78.30]% | 94.85 [93.44, 96.26]% |
| Adv AUROC ↑ | 83.67 [81.38, 85.95]% | 81.81 [79.13, 84.48]% | 88.73 [86.94, 90.53]% | 90.26 [89.61, 90.92]% | 83.43 [71.25, 95.62]% | 53.56 [44.58, 62.54]% | 53.62 [46.89, 60.36]% | 95.72 [94.71, 96.74]% |
| | | | | Maximum Probability | | | | |
| ID Acc ↑ | 99.76 [99.70, 99.83]% | 99.81 [99.74, 99.87]% | 99.09 [97.78, 100.00]% | 99.92 [99.89, 99.95]% | 99.80 [99.67, 99.92]% | 99.70 [99.54, 99.85]% | 99.62 [99.52, 99.73]% | 99.92 [99.87, 99.97]% |
| ID Cov ↑ | 81.59 [78.82, 84.36]% | 77.70 [67.89, 87.50]% | 80.47 [75.12, 85.82]% | 78.55 [74.16, 82.95]% | 87.21 [82.29, 92.14]% | 87.82 [83.01, 92.63]% | 81.34 [75.68, 87.00]% | 90.47 [89.25, 91.70]% |
| OOD Cov ↓ | 20.78 [13.03, 28.54]% | 16.77 [10.96, 22.58]% | 23.01 [6.50, 39.51]% | 20.67 [12.82, 28.52]% | 36.60 [13.52, 59.68]% | 29.76 [14.28, 45.24]% | 38.39 [19.61, 57.17]% | 6.77 [4.40, 9.14]% |
| Adv Cov ↓ | 26.39 [17.51, 35.27]% | 21.49 [17.41, 25.57]% | 14.46 [0.00, 32.75]% | 15.23 [6.93, 23.52]% | 67.96 [56.71, 79.22]% | 49.46 [38.11, 60.81]% | 52.31 [39.03, 65.60]% | 3.34 [1.46, 5.23]% |
| AUROC ↑ | 87.07 [82.92, 91.22]% | 87.59 [85.42, 89.75]% | 85.17 [77.80, 92.55]% | 85.18 [81.08, 89.28]% | 75.54 [57.27, 93.81]% | 82.09 [71.73, 92.45]% | 75.28 [62.80, 87.75]% | 96.59 [95.44, 97.74]% |
| Adv AUROC ↑ | 84.49 [80.63, 88.35]% | 83.66 [79.94, 87.37]% | 89.42 [81.23, 97.60]% | 87.99 [84.67, 91.30]% | 50.24 [39.88, 60.60]% | 61.29 [49.92, 72.66]% | 63.06 [52.00, 74.12]% | 97.94 [97.07, 98.80]% |
| | | | | Differential Entropy | | | | |
| ID Acc ↑ | - | - | 96.86 [95.04, 98.68]% | - | 99.69 [99.57, 99.82]% | 99.67 [99.54, 99.79]% | 99.39 [99.27, 99.52]% | 99.82 [99.74, 99.89]% |
| ID Cov ↑ | - | - | 73.62 [69.65, 77.60]% | - | 86.82 [80.73, 92.91]% | 88.13 [84.83, 91.44]% | 79.31 [74.29, 84.34]% | 91.55 [90.13, 92.97]% |
| OOD Cov ↓ | - | - | 34.44 [24.43, 44.44]% | - | 34.86 [12.62, 57.10]% | 39.13 [9.16, 69.10]% | 39.91 [32.67, 47.16]% | 7.83 [5.62, 10.03]% |
| Adv Cov ↓ | - | - | 31.24 [22.42, 40.06]% | - | 60.45 [36.50, 84.41]% | 65.68 [57.08, 74.28]% | 58.22 [41.35, 75.09]% | 3.30 [1.56, 5.04]% |
| AUROC ↑ | - | - | 73.40 [66.66, 80.13]% | - | 77.03 [62.33, 91.72]% | 73.09 [47.84, 98.34]% | 74.25 [72.14, 76.37]% | 96.89 [96.03, 97.75]% |
| Adv AUROC ↑ | - | - | 74.82 [67.42, 82.22]% | - | 52.98 [32.28, 73.68]% | 52.85 [45.34, 60.35]% | 59.30 [44.23, 74.37]% | 98.08 [97.75, 98.40]% |

A similar trend emerges under FGSM perturbations. While Whitebox attains strong performance at moderate to high perturbations ($\epsilon = 0.5, 1.0$), GUIDE maintains consistently competitive results across the entire range, outperforming all alternatives at lower perturbation strengths where robustness is most challenging. Importantly, GUIDE's performance remains within narrow confidence intervals, underscoring both its stability and reliability in adversarial settings.

For the non-gradient-based Salt & Pepper noise attack, GUIDE again dominates across all perturbation magnitudes, exceeding 96% AUROC throughout. Competing baselines such as ABNN and Whitebox achieve high values in isolated cases, but their performance is less stable and fails to consistently match GUIDE's superior robustness. The fact that GUIDE maintains high AUROC scores across gradient-based and non-gradient-based attacks alike highlights its versatility and generalizability. Taken together, these results demonstrate that GUIDE is the most effective method among the evaluated approaches.

## B.3 THRESHOLD ANALYSIS

Table 5 reports the performance of comparative approaches across a variety of uncertainty metrics used for the scoring rule. The results demonstrate that GUIDE achieves state-of-the-art performance consistently across all metrics, while alternative methods show clear weaknesses in either calibration, coverage, or robustness.

Under the Mutual Information metric, all methods achieve high in-distribution accuracy (above 99%), with Whitebox reaching the highest overall accuracy at 99.92%. However, accuracy differences at this scale are marginal and do not reflect robustness under distributional shift. For example, while EMM + cal attains the best in-distribution coverage (92.32%), it fails dramatically on adversarial coverage, yielding 55.47% compared to just 4.60% for GUIDE. Similarly, Whitebox reduces adversarial coverage to 9.04%, but GUIDE achieves the best overall balance by combining strong ID coverage (87.05%) with the lowest OOD (7.34%) and adversarial coverage (4.60%). These values translate into AUROC scores of 94.85% for OOD detection and 95.72% for adversarial detection, both substantially ahead of the next-best method.

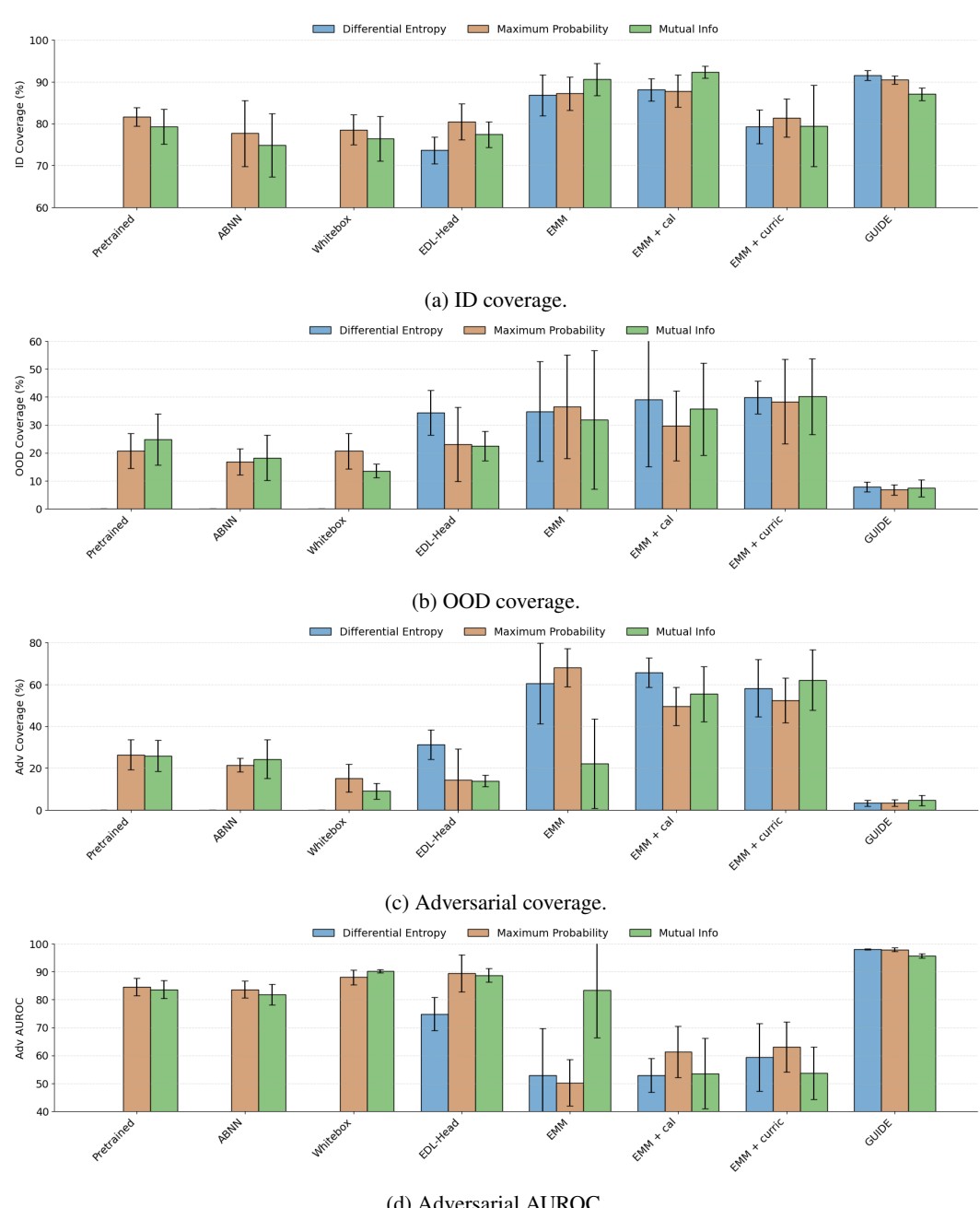

(a) ID coverage.

(b) OOD coverage.

(c) Adversarial coverage.

(d) Adversarial AUROC.

Figure 13: Evaluated metrics for different ID-OOD threshold metrics (differential entropy, maximum probability, mutual information) where the ID dataset is MNIST, and the OOD dataset is FashionM-NIST.

With the Maximum Probability metric, GUIDE again delivers the strongest results across nearly all evaluation criteria. It matches Whitebox in in-distribution accuracy at 99.92% but provides markedly improved robustness. For instance, OOD coverage is reduced to 6.77% and adversarial coverage to only 3.34%, whereas the best competing approaches (ABNN and Whitebox) still remain above 15%. This robustness translates into AUROC scores of 96.59% for OOD detection and 97.94% for adversarial detection, again setting the clear benchmark.

GUIDE's robust performance is most evident under Differential Entropy, where it achieves near-perfect separation of ID and shifted samples. Specifically, GUIDE attains an ID coverage of 91.55%,

Table 6: The accuracy, OOD detection, and adversarial attack detection, AUROC, and adversarial AUROC performance of GUIDE under different values of the $T$ hyperparameter. The adversarial attack is L2PGD (1.0 maximum perturbation), and the ID and OOD datasets are MNIST and FashionMNIST.

| | 2.0 | 5.0 | 10.0 | 20.0 |
|---|---|---|---|---|
| ID Acc ↑ | 99.87 [99.80, 99.94]% | 99.87 [99.82, 99.92]% | 99.70 [99.59, 99.81]% | 99.75 [99.63, 99.87]% |
| ID Cov ↑ | 88.21 [85.14, 91.28]% | 87.05 [85.24, 88.85]% | 89.62 [86.81, 92.44]% | 87.78 [83.51, 92.04]% |
| OOD Cov ↓ | 9.12 [6.62, 11.62]% | 7.34 [3.54, 11.14]% | 7.84 [5.08, 10.60]% | 7.73 [3.30, 12.16]% |
| Adv Cov ↓ | 9.42 [5.67, 13.17]% | 4.60 [1.61, 7.58]% | 8.98 [5.38, 12.59]% | 6.67 [4.92, 8.42]% |
| AUROC ↑ | 94.50 [94.14, 94.86]% | 94.85 [93.44, 96.26]% | 95.24 [94.21, 96.28]% | 94.51 [92.84, 96.18]% |
| Adv AUROC ↑ | 94.60 [93.73, 95.48]% | 95.72 [94.71, 96.74]% | 95.04 [94.44, 95.64]% | 94.58 [92.64, 96.52]% |

Table 7: The accuracy, OOD detection, and adversarial attack detection, AUROC, and adversarial AUROC performance of GUIDE under different values of the $\gamma$ hyperparameter. The adversarial attack is L2PGD (1.0 maximum perturbation), and the ID and OOD datasets are MNIST and FashionMNIST.

| | 0.10 | 0.25 | 0.50 | 0.75 | 1.00 |
|---|---|---|---|---|---|
| ID Acc ↑ | 99.88 [99.80, 99.96]% | 99.87 [99.82, 99.92]% | 99.82 [99.74, 99.89]% | 99.84 [99.72, 99.95]% | 99.76 [99.70, 99.82]% |
| ID Cov ↑ | 87.82 [83.45, 92.19]% | 87.05 [85.24, 88.85]% | 88.31 [84.30, 92.31]% | 87.13 [83.28, 90.99]% | 91.69 [90.50, 92.88]% |
| OOD Cov ↓ | 9.87 [6.07, 13.67]% | 7.34 [3.54, 11.14]% | 8.92 [6.78, 11.06]% | 13.34 [11.30, 15.38]% | 12.14 [8.48, 15.80]% |
| Adv Cov ↓ | 8.73 [4.70, 12.76]% | 4.60 [1.61, 7.58]% | 9.79 [4.26, 15.31]% | 6.49 [3.08, 9.91]% | 13.37 [8.27, 18.47]% |
| AUROC ↑ | 93.78 [90.60, 96.97]% | 94.85 [93.44, 96.26]% | 94.41 [92.16, 96.67]% | 91.52 [89.62, 93.42]% | 94.28 [91.94, 96.62]% |
| Adv AUROC ↑ | 94.51 [92.67, 96.35]% | 95.72 [94.71, 96.74]% | 93.54 [91.17, 95.92]% | 94.48 [92.82, 96.14]% | 93.59 [90.76, 96.41]% |

OOD coverage as low as 7.83%, and adversarial coverage of just 3.30%. The corresponding AUROC scores are 96.89% for OOD detection and 98.08% for adversarial detection, surpassing all baselines by wide margins. In contrast, alternative methods such as EDL-Head or EMM variants struggle, with adversarial AUROC values often below 75%. Taken together, these results show that although some baselines excel in isolated metrics (e.g., Whitebox in raw ID accuracy or EMM + cal in ID coverage), only GUIDE achieves consistently strong and balanced performance across ID, OOD, and adversarial evaluations. These results can also be visualised in Figure 13 for further comparative analysis.

## B.4 ABLATION ANALYSIS

Table 6 presents the effect of varying the hyperparameter $T$ on GUIDE's performance. Across all values, in-distribution accuracy remains very high (above 99.7%), confirming that predictive performance is stable with respect to $T$. The most notable differences arise in coverage and adversarial robustness. At $T = 2$, adversarial coverage is relatively higher (9.42%), while at $T = 5$ it is minimised to 4.60%, with a corresponding adversarial AUROC of 95.72%, the strongest result observed. Larger values of $T$ (10 and 20) slightly improve AUROC on OOD detection (up to 95.24%) but come with modest increases in adversarial coverage (around 7-9%).

Table 7 examines the sensitivity of GUIDE to the weighting parameter $\gamma$. In-distribution accuracy is uniformly high across settings ($\approx 99.8\%$), indicating that predictive accuracy is largely insensitive to $\gamma$. The primary effects manifest in coverage and robustness: a moderate value, $\gamma = 0.25$, offers the most balanced performance, attaining the lowest adversarial coverage (4.60%) with strong OOD coverage (7.34%) and the highest AUROC/Adv-AUROC pair (94.85%/95.72%). Lowering $\gamma$ to 0.10 slightly increases both OOD and adversarial coverage (9.87% and 8.73%) and reduces OOD AUROC to 93.78%. Increasing $\gamma$ beyond 0.25 raises ID coverage (e.g., 91.69% at $\gamma = 1.00$) but at the cost of markedly higher spurious coverage on OOD and adversarial inputs (12.14% and 13.37%, respectively) and weaker robustness (Adv-AUROC 93.59%). The degradation is most apparent at $\gamma = 0.75$, where OOD coverage peaks at 13.34% and OOD AUROC dips to 91.52%. Overall, the results suggest that creating a monotonic slope that is too sharp could degrade performance, but very slightly.

Finally, Table 8 analyses the impact of varying the $\eta$ hyperparameter on GUIDE. Across all values, in-distribution accuracy remains consistently high ($\approx 99.8$–99.9%), indicating that predictive performance is largely unaffected by $\eta$. The main differences emerge in coverage and computational cost. Moderate settings ($\eta = 0.9$) achieve the strongest balance, with adversarial coverage reduced to

Table 8: The accuracy, OOD detection, and adversarial attack detection, AUROC, and adversarial AUROC performance of GUIDE under different values of the $\eta$ hyperparameter. Also included are the selected layers for each $\eta$, and the training and inference time, in seconds, to train and infer upon the whole dataset. The adversarial attack is L2PGD (1.0 maximum perturbation), and the ID and OOD datasets are MNIST and FashionMNIST.

| | 0.50 | 0.60 | 0.70 | 0.80 | 0.90 | 1.00 |
|---|---|---|---|---|---|---|
| ID Acc ↑ | 99.78 [99.69, 99.87]% | 99.84 [99.77, 99.91]% | 99.78 [99.67, 99.88]% | 99.83 [99.74, 99.91]% | 99.87 [99.82, 99.92]% | 99.90 [99.87, 99.94]% |
| ID Cov ↑ | 88.21 [85.74, 90.69]% | 85.04 [83.33, 86.74]% | 87.67 [83.65, 91.68]% | 86.65 [82.69, 90.61]% | 87.05 [85.24, 88.85]% | 87.73 [83.68, 91.78]% |
| OOD Cov ↓ | 7.40 [4.19, 10.61]% | 7.86 [4.92, 10.80]% | 6.21 [2.80, 9.62]% | 7.41 [6.16, 8.66]% | 7.34 [3.54, 11.14]% | 6.64 [5.25, 8.04]% |
| Adv Cov ↓ | 5.81 [3.05, 8.57]% | 5.86 [4.47, 7.25]% | 5.95 [4.27, 7.63]% | 5.37 [3.90, 6.84]% | 4.60 [1.61, 7.58]% | 7.71 [4.55, 10.87]% |
| AUROC ↑ | 94.82 [92.58, 97.05]% | 93.89 [92.59, 95.19]% | 95.32 [93.90, 96.73]% | 94.35 [92.85, 95.85]% | 94.85 [93.44, 96.26]% | 95.11 [93.52, 96.70]% |
| Adv AUROC ↑ | 95.45 [94.02, 96.87]% | 94.29 [93.60, 94.98]% | 95.14 [93.87, 96.41]% | 94.43 [92.07, 96.78]% | 95.72 [94.71, 96.74]% | 94.64 [93.10, 96.18]% |
| Selected Layers | {pool1} | {pool1, conv1} | {pool1, conv1} | {pool1, conv1} | {pool1, conv1, pool2} | {pool1, conv1, pool2, conv2} |
| Training Time (s) | 287.87 [285.75, 289.99] | 285.83 [279.56, 292.10] | 289.87 [286.02, 293.72] | 287.56 [282.50, 292.61] | 288.45 [283.93, 292.98] | 295.06 [290.60, 299.51] |
| Inference Time (s) | 1.66 [1.12, 2.20] | 1.77 [1.27, 2.27] | 1.65 [1.09, 2.21] | 1.68 [1.16, 2.19] | 1.94 [1.33, 2.56] | 2.38 [1.63, 3.12] |

4.60% and corresponding adversarial AUROC of 95.72%, while also maintaining competitive OOD coverage (7.34%). Lower $\eta$ values (e.g., 0.5-0.7) yield similar AUROC (around 95%) but involve fewer selected layers and slightly lower inference times ($\sim$1.65s). In contrast, higher $\eta$ values (1.0) expand the set of selected layers ({pool1, conv1, pool2, conv2}), which increases inference time to 2.38s and slightly worsens adversarial coverage (7.71%). This is due to there being more gradients for a gradient-based attack to fool.

## C EXPERIMENT DETAILS

This section outlines our experimental setup in detail to ensure reproducibility and to make the evaluation protocol transparent. We first present the datasets employed for both in-distribution (ID) and out-of-distribution (OOD) evaluation, after which we describe the threshold-based scoring metrics used for uncertainty-driven rejection. Next, we summarise the comparative methods considered in our study and provide a thorough account of the model training process. Lastly, we report task-specific configurations and implementation details pertinent to each experimental setting.

### C.1 DATASETS

We conduct evaluations using a broad suite of well-established computer vision benchmarks that cover different domains and levels of task complexity. This design allows for a thorough assessment of robustness and uncertainty estimation across diverse conditions. To examine the capacity of models to discriminate between in-distribution (ID) and out-of-distribution (OOD) samples, we consider both near-OOD and far-OOD scenarios. The near-OOD case involves datasets with partial class overlap relative to the ID data, whereas far-OOD datasets originate from entirely distinct visual domains. Detailed information on each dataset, including sample counts, input resolution, class composition, and data splits, is provided below. Representative examples are displayed in Figure 14.

- **MNIST LeCun et al. (1998):** consists of 28x28 greyscale images of handwritten digits (0-9) spanning 10 classes. We utilise the widely used standard split of 60,000 training samples, 8000 test samples, and 2000 validation samples. Pixel normalisation was applied, bounded [0, 1]. In our experimental evaluation, MNIST serves as one of the ID datasets. MNIST was paired with FashionMNIST and KMNIST as far-OOD datasets due to them sharing no overlapping classes, but exhibit similar visual characteristics, including greyscale appearance, low resolution, and hand-drawn style. MNIST was also paired with EMNIST as a near-OOD dataset due to it sharing the same visual characteristics and sharing overlapping classes with EMNIST containing handwritten digits in addition to handwritten letters.
- **FashionMNIST Xiao et al. (2017):** consists of 28x28 greyscale images of clothing items (e.g., coats, bags, t-shirts, etc.) spanning 10 classes. We utilise the widely used standard split of 60,000 training sample, 8000 test samples, and 2000 validation samples. Pixel normalisation was applied, bounded [0, 1]. In our experimental evaluation, we use this dataset as a far-OOD pairing with the ID MNIST dataset.
- **KMNIST Clanuwat et al. (2018):** consists of 28x28 greyscale images of handwritten Japanese characters (specifically Kuzushiji characters from classical Japanese literature) spanning 10 classes. We utilise the widely used standard split of 60,000 training samples, 8000 test samples, and 2000 validation samples. Pixel normalisation was applied, bounded

$[0, 1]$. In our experimental evaluation, we use this dataset as a far-OOD pairing with the ID MNIST dataset.

- **EMNIST Cohen et al. (2017):** consists of 28x28 greyscale images of both handwritten letters (from the Latin alphabet) and digits (0-9) spanning 47 classes. We use a split of 112,800 training samples, 15,040 test samples, and 3760 validation samples. Pixel normalisation was applied, bounded $[0, 1]$. In our experimental evaluation, we use this dataset as a near-OOD pairing with the ID MNIST dataset due to it sharing the same visual characteristics and overlapping classes with MNIST (handwritten digits).
- **CIFAR10 Krizhevsky et al. (2009):** consists of 32x32 RGB images of real-world objects (e.g, birds, trucks, airplanes, frogs, etc) spanning 10 classes. We utilise the widely used standard split of 50,000 training samples, 8000 test samples, and 2000 validation samples. Pixel normalisation was applied, bounded $[0, 1]$ across all three RGB channels. In our experimental evaluation, CIFAR10 serves as one of the ID datasets. CIFAR10 was paired with FashionMNIST and SVHN as a far-OOD dataset due to them sharing no overlapping classes, but exhibiting similar visual characteristics, including coloured appearance, real-world pictures. CIFAR-10 was also paired with CIFAR-100 as a near-OOD dataset, due to the datasets sharing the same visual characteristics and overlapping classes. Specifically, CIFAR-10 contains broad object categories (e.g., cat, dog, truck, ship), which correspond to finer-grained classes like (e.g., house cat, beagle, pickup truck, cruise ship) in CIFAR-100.
- **CIFAR100 Krizhevsky et al. (2009):** consists of 32x32 RGB images of real-world objects spanning of 100 classes that are fine-grained counterparts of the classes from CIFAR10. For example, CIFAR10 has the class *truck*, whilst CIFAR100 has the classes *pickup truck* and *train*. We utilise the widely used standard split of 50,000 training samples, 8000 test samples, and 2000 validation samples. Pixel normalisation was applied, bounded $[0, 1]$ across all three RGB channels. In our experimental evaluation, we use this dataset as a near-OOD pairing with the ID CIFAR10 dataset due to it sharing the same visual characteristics and overlapping classes.
- **SVHN Netzer et al. (2011):** consists of 32x32 RGB images of house numbers collected from Google Street View spanning 10 classes (0-9). We use a split of 73,257 training samples, 10,832 test samples, and 5200 validation samples. Pixel normalisation was applied, bounded $[0, 1]$ across all three RGB channels. In our experimental evaluation, we use this dataset as a far-OOD pairing with the ID CIFAR10 dataset due to the substantial domain shift between street-level digit photographs and object-centric natural images.
- **Oxford Flowers Nilsback & Zisserman (2008):** consists of 64x64 RGB images of flowers commonly found in the United Kingdom, spanning 102 (e.g., Water Lily, Wild Pansy, etc) classes. We utilise a split of 9826 training samples, 1638 test samples, and 1638 validation samples. This is double the size of the original dataset, as each image has been duplicated with a random augmentation to aid generalisation within training. Pixel normalisation was applied, bounded $[0, 1]$ across all three RGB channels. In our experimental evaluation, we use this dataset as an ID dataset. Oxford Flowers was paired with Deep Weeds because they share no overlapping classes but exhibit similar visual characteristics, including features found in nature (leaves, flowers, etc.).
- **Deep Weeds Olsen et al. (2019):** consists of 64x64 RGB images of species of weeds found in Australia, spanning over 8 classes (e.g., Snake weed, Rubber vine, etc). We use the standard split of 10,505 training samples, 3502 test samples, and 3502 validation samples. Pixel normalisation was applied, bounded $[0, 1]$ across all three RGB channels. In our experimental evaluation, Deep Weeds serves as a far-OOD pairing with the ID Oxford Flowers dataset due to the substantial domain shift between photographs of flowers and photographs of weeds.

For datasets lacking predefined partitions, we created manual splits using stratification to maintain consistent class distributions across subsets. In all experiments, the validation portion of each dataset was employed to calibrate the ID–OOD threshold for uncertainty-based rejection. This calibration enabled evaluation of both ID and OOD coverage on the corresponding test sets.

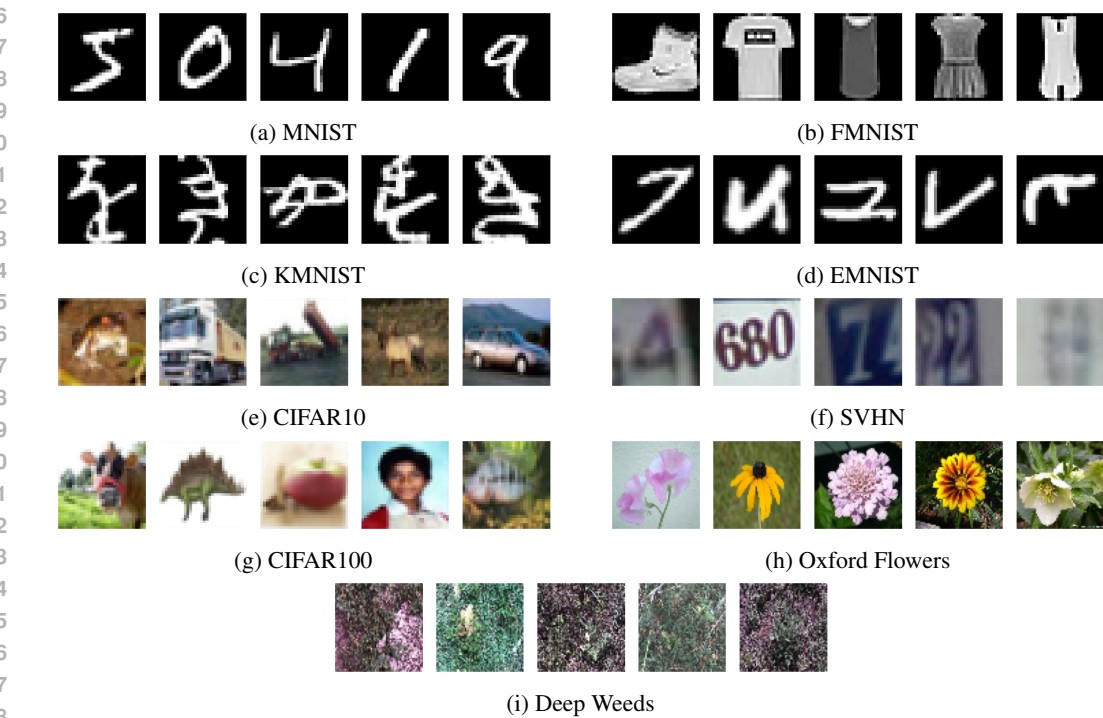

(a) MNIST          (b) FMNIST

(c) KMNIST          (d) EMNIST

(e) CIFAR10          (f) SVHN

(g) CIFAR100          (h) Oxford Flowers

(i) Deep Weeds

Figure 14: Example images from the datasets used in our experimental evaluation.

## C.2 ID-OOD Thresholds

To support abstention from uncertain predictions, we determine the optimal ID–OOD threshold using the validation set corresponding to each experimental configuration. Our evaluation considers several ID–OOD scoring metrics, enabling a rigorous assessment of GUIDE alongside the baseline methods.

- **Differential Entropy:** measures the dispersion or uncertainty inherent in the Dirichlet distribution. It is given by:

$$\sum_{k=1}^{K} \ln \Gamma(\alpha_k) - \ln \Gamma(S) - \sum_{k=1}^{K} (\alpha_k - 1)(\Psi(\alpha_k) - \Psi(S))$$

where $B(\boldsymbol{\alpha})$ represents the multivariate Beta function, $S$ is the overall concentration parameter, and $\Psi(\cdot)$ is the digamma function. Larger entropy values indicate greater uncertainty, which is typically observed for OOD inputs. This score applies only to Dirichlet-based uncertainty models (e.g., posterior networks, evidential networks).

- **Mutual Information:** captures epistemic uncertainty by quantifying the amount of information that the model parameters contribute to the predictive distribution. It is expressed as the gap between the predictive entropy and the expected conditional entropy under the posterior over parameters:

$$-H\big(\mathbb{E}_{q(\omega)}[\,p(y \mid x, \omega)\,]\big) \;-\; \mathbb{E}_{q(\omega)}[\,H(p(y \mid x, \omega))]$$

where $H(p) = -\sum_{k=1}^{K} p_k \log p_k$ and $q(\omega)$ is the (explicit or implicit) posterior over parameters.

- **Maximum Probability:** is a simple confidence score that measures the model's most likely prediction. It is defined as the maximum softmax probability over all classes:

$$\max_{k \in \{1, \ldots, K\}} \; p(y = k \mid x)$$

Higher values indicate that the model is more confident in its predicted class, while lower values suggest greater uncertainty.

In line with prior works Shen et al. (2023), the optimal ID–OOD threshold is determined using the validation datasets. For a selected scoring metric (chosen from the list above), we first compute scores on both ID and OOD validation samples. These scores are then used to construct a receiver operating

characteristic (ROC) curve, where ID samples are regarded as positive and OOD samples as negative. The ROC curve provides a visualisation of how well the chosen metric separates ID from OOD data. To identify the decision boundary, we calculate the threshold that maximises TPR − FPR, yielding the point of optimal separation. This criterion achieves a principled balance between retaining ID inputs and filtering out OOD inputs, avoiding the need for ad hoc tuning, and enabling clear reporting of ID and OOD coverage under deployment-like conditions.

After this calibration stage, the threshold is fixed and subsequently applied to the test sets. During evaluation, any test input whose score (under the chosen metric) crosses this threshold is rejected. This provides a consistent basis for comparing ID retention and OOD rejection across all models and datasets.

### C.3 COMPARATIVE APPROACHES

To assess the performance of GUIDE, we benchmark it against a set of recent post-hoc uncertainty estimation methods that capture the state of the art in this area. Below, we provide a concise description of each method together with the implementation details and hyperparameter settings adopted in our experiments:

- **Standard Neural Network (Pretrained):** serves as a baseline classifier without an explicit uncertainty modelling component. The model is trained using conventional cross-entropy loss on the in-distribution training data, and outputs softmax probabilities over the class set.
- **Adaptable Bayesian Neural Networks (ABNN) Franchi et al. (2024):** provide a lightweight post-hoc strategy for equipping pretrained deterministic networks with Bayesian uncertainty estimation capabilities. Rather than retraining the full model or relying on costly ensembles, ABNN introduces Bayesian Normalization Layers (BNLs), which inject Gaussian perturbations into existing normalization layers. Only the parameters of these BNLs are fine-tuned for a few epochs, enabling the deterministic DNN to be transformed into a scalable approximation of a BNN with minimal overhead. During inference, multiple stochastic forward passes are performed by sampling perturbations, producing diverse predictions that capture epistemic uncertainty.
- **Whitebox Chen et al. (2019):** is a post-hoc meta-model based confidence scoring approach that observes the internal representations of a base classifier. It introduces linear classifier probes at multiple intermediate layers of the base model, and trains an auxiliary meta-model on the probe outputs to predict whether the base model's prediction is correct.
- **EDL-Head Sensoy et al. (2018):** represents a simple evidential deep learning baseline built directly on top of a pretrained classifier. The softmax output layer of the pretrained model is removed and replaced with an evidential head that predicts the Dirichlet concentration parameters $\alpha$ for each class. The evidential head is then trained while keeping the backbone fixed, allowing the model to encode both aleatoric and epistemic uncertainty through the resulting Dirichlet distribution. This setup provides a direct comparison to standard softmax classifiers by isolating the effect of evidential parameterisation on uncertainty estimation.
- **Dirichlet Meta-Model (EMM) Shen et al. (2023):** is a post-hoc uncertainty learning method that introduces a meta-model that leverages intermediate feature representations from the base model and parameterises a Dirichlet distribution over class probabilities. By training only this lightweight meta-model, EMM captures both aleatoric and epistemic uncertainty while leaving the predictive performance of the base model unchanged. Training is performed using an ELBO loss that balances classification accuracy with uncertainty calibration, regularised by a KL-divergence term to avoid overconfidence. During inference, the Dirichlet meta-model produces concentration parameters whose dispersion reflects uncertainty, enabling effective applications to OOD detection, misclassification detection, and trustworthy transfer learning.
- **EMM + Cal:** extends the Dirichlet Meta-Model (EMM) Shen et al. (2023) with the saliency-based calibration stage from GUIDE. Instead of arbitrarily selecting intermediate layers for the meta-model, this variant uses relevance propagation to identify the most salient layers of the pretrained backbone, ensuring that the evidential head is trained on semantically meaningful features.
- **EMM + Curric:** modifies the Dirichlet Meta-Model (EMM) Shen et al. (2023) by incorporating only the uncertainty-guided curriculum component from GUIDE, while omitting the saliency-based calibration stage and the new loss formulation. In this setup, the evidential

head is trained on arbitrarily chosen intermediate features, but training proceeds under a monotonic noise-driven curriculum that gradually corrupts inputs from low to high noise levels.

## C.4 ADVERSARIAL ATTACKS

This section describes the adversarial attack strategies employed to evaluate the robustness of uncertainty-aware models. Adversarial attacks consist of carefully designed, imperceptible perturbations that can cause models to misclassify while still producing highly confident predictions. Such manipulations pose a particular challenge for uncertainty-based systems, as they can distort both aleatoric and epistemic signals, leading adversarial inputs to resemble in-distribution samples or bypass detection mechanisms.

To ensure a thorough robustness evaluation, we examine both gradient-driven and gradient-free attacks. Gradient-based methods, characteristic of white-box settings, exploit direct access to model gradients in order to craft adversarial perturbations. In contrast, gradient-free approaches inject random or structured noise into inputs and are more representative of black-box adversarial scenarios. All attacks in our study are implemented using Foolbox Rauber et al. (2017), covering the following configurations:

- **L2 Projected Gradient Descent (L2PGD):** is an iterative, white-box adversarial attack that perturbs inputs within a bounded $L_2$-norm ball to maximise the model's loss. At each iteration, the input is updated in the direction of the gradient of the loss with respect to the input, followed by projection back onto the $L_2$-ball of radius $\epsilon$. This results in smooth, high-precision perturbations that remain less perceptible to humans:

$$x'_{t+1} = \text{Proj}_\epsilon^{L_2}(x'_t + \alpha \cdot \nabla_x J(x'_t, y))$$

  where $\alpha$ is the step size and $J(x, y)$ is the loss function.
- **Fast Gradient Sign Method (FGSM):** is a single-step white-box adversarial attack that perturbs the input in the direction of the sign of the gradient of the loss:

$$x' = x + \epsilon \cdot \text{sign}(\nabla_x J(x, y))$$

  where $\epsilon$ controls the perturbation magnitude. FGSM generates perceptible but targeted perturbations with minimal computational overhead.
- **Salt & Pepper Noise:** is a non-gradient-based black-box perturbation that randomly sets a proportion of input pixels to their minimum or maximum value. This form of structured noise simulates impulsive corruption and tests the model's resilience to sparse, high-intensity artefacts. It does not rely on model gradients and is agnostic to internal model parameters.

## C.5 TRAINING DETAILS

To ensure consistency and reproducibility across all experiments, we adopt a fixed training configuration and architectural setup for all models, including our proposed GUIDE variants and baseline comparators.

All models were trained using a custom convolutional neural network architectures. Table 9 shows the architecture of these models. LeNet-5 was used for MNIST dataset pairs, ResNet-20 was used for CIFAR-10 → SVHN, and SE-Net was used for CIFAR-10 → CIFAR-100 and Oxford Flowers dataset pairs. The output layer of every model consists of $K$ units (one per class according to the respective ID dataset). All models are trained using the Adam optimiser Kingma & Ba (2014).

All experiments were implemented in TensorFlow 2.15 and executed on a large performance GPU cluster using a maximum of three Nvidia A40 GPUs, 32 CPU cores, 167GB of memory (per GPU). All models were trained from scratch, and no pretraining or external data was used. All runs used random seeds.

Under our parameterisation, the GUIDE loss is continuously differentiable with respect to the meta-model parameters $\phi$: the map $\phi \mapsto \alpha(x)$ is smooth (affine layers composed with $\exp$ ensure $\alpha_k > 0$), the Dirichlet-ELBO terms involve $C^\infty$ functions (gamma/digamma), and the SRE component is a smooth rational function of $\alpha$. If non-smooth activations (e.g., ReLU) are used internally, the GUIDE loss is differentiable almost everywhere, and we rely on subgradients, as is standard in first-order optimisation.

Table 9: Architectures and hyperparameters of the base networks used within our experimental evaluation. Conv$(C, k, s)$ = $k \times k$ conv, $C$ channels, stride $s$; MP/AP = max/avg pool ($2 \times 2$, $s{=}2$); BN = batch norm; GAP = global avg pool; SE$(r)$ = squeeze–excitation (reduction $r$); BB$(C, s)$ = basic ResNet block with two $3 \times 3$ convs, projection skip if $s{=}2$.

|  | LeNet-5 | ResNet-20 | SE-Net |
|---|---|---|---|
| Input | $H \times W \times D$ | $H \times W \times D$ | $H \times W \times D$ |
| Stage 1 | Conv(6, 5, 1), tanh, *pad=same* $\to$ AP [pool1] | Conv(16, 3, 1), *pad=same*, no bias $\to$ BN $\to$ ReLU | Conv(32, 3, 1) + ReLU $\to$ BN $\to$ SE(8) $\to$ MP [pool1] |
| Stage 2 | Conv(16, 5, 1), tanh, *valid* $\to$ AP [pool2] | $3\times$ BB(16, 1) | Conv(64, 3, 1) + ReLU $\to$ BN $\to$ SE(8) $\to$ MP [pool2] |
| Stage 3 | Conv(120, 5, 1), tanh, *valid* $\to$ Flatten | BB(32, **2**) + $2\times$ BB(32, 1) | Conv(128, 3, 1) + ReLU $\to$ BN $\to$ SE(8) $\to$ GAP |
| Head | FC(84), tanh $\to$ FC($K$), softmax | BB(64, **2**) + $2\times$ BB(64, 1) $\to$ GAP $\to$ FC($K$), softmax | FC(128) $\to$ Dropout(0.5) $\to$ FC($K$), softmax |
| Epochs | 50 | 200 | 200 |
| KL-Weight ($\lambda_{kl}$) | 1e-1 | 1e-3 | 1e-3 |
| Prior ($\beta$) | 1.0 | 1.0 | 1.0 |
| Learning rate | 1e-2 | 1e-2 | 1e-4 |

## C.6 EXPERIMENTAL SETUPS

This subsection outlines additional implementation details and experimental choices that are specific to certain experiments that may help with reproducibility and transparency.

For practical stability, we restrict candidate layers to semantically meaningful, deterministic representations (Conv/Linear, Norm in inference mode, Pool/Flatten, and post-activation block outputs). We exclude stochastic layers (Dropout/Noise), non-differentiable or discrete operators (e.g., Argmax/Top-k/Quantize implemented via Lambda), softmax activations, and element-wise merge/gate nodes (Add/Multiply/Min/Max/Average). All relevance and feature extraction are computed with the backbone in inference mode to avoid perturbing internal statistics.

In particular, for architectures with explicit residual or modular stage boundaries (e.g., ResNet, WideResNet), we consider only post-activation outputs at the end of each residual block or stage (after the skip connection addition and final ReLU), as these represent the actual semantic feature maps propagated forward in the backbone. For non-residual models (e.g., VGG, LeNet, SENet), we take stage-level post-activation features after pooling layers, global average pooling, or fully connected layers, as appropriate. This ensures that all tapped representations are stable, comparable in scale, and maximally informative for the meta-model.

In Table 1, the adversarial attack used is a gradient-based L2PGD attack and the uncertainty metric was mutual information. For MNIST to FashionMNIST, KMNIST, and EMNIST, a maximum perturbation of 1.0 was used. For CIFAR10 to SVHN, CIFAR10 to CIFAR100, and Oxford Flowers to Deep Weeds, a maximum perturbation of 0.1 was used. This choice reflects the fact that adversarial examples on natural image datasets such as CIFAR-10 require smaller perturbations to significantly degrade model performance, due to the increased complexity and lower inherent separability of the visual features. In contrast, MNIST-like datasets typically require larger perturbations to achieve a comparable adversarial effect.

For all experiments unless stated otherwise, GUIDE and its variants utilises the hyperparameter combination of $T = 5$, $\gamma = 0.25$, and $\eta = 0.9$ for MNIST to FashionMNIST, KMNIST, and EMNIST. A combination of $T = 10$, $\gamma = 0.05$, and $\eta = 0.9$ for CIFAR10 to SVHN, CIFAR10 to CIFAR100, and Oxford Flowers to Deep Weeds.

