# OpenReview forum: "Learning When to Be Uncertain: A Post-Hoc Meta-Model for Guided Uncertainty Learning"
_ICLR.cc/2026/Conference — Submitted to ICLR 2026_

### Official Review · Reviewer_7wVu · 2025-10-29

**Soundness:** 2
**Presentation:** 2
**Contribution:** 2
**Rating:** 4
**Confidence:** 3

**Summary:**

The paper introduces GUIDE (Guided Uncertainty Learning Using a Post-hoc Evidential Meta-model), a post-hoc non intrusive method for uncertainty quantification that can be applied to pretrained deterministic neural networks without retraining them.

**Strengths:**

- The paper is easy to read
- The topic can be relevant, and the approach is interesting

**Weaknesses:**

**Motivation and definition of post-hoc uncertainty:** The motivation behind the paper, as per the abstract, is that existing post-hoc approaches inherit “misplaced” confidence, or reshape the predictions (through temperature scaling for instance, I believe). However, these motivations are not being developed and explained in the later sections of the manuscript, which makes the rationale behind GUIDE somehow unclear. While I just did not like that, since I believe it was not so relevant after all, I then noticed that the authors' definition of post-hoc is a bit ambiguous. By reading, one is left believing that the definition refers to approaches acting on a pretrained network. However, the authors mention as post-hoc method the one proposed in [int_ref_1], which do performs end-to-end training. This makes the whole paper framing confusing, and makes the lack of description of limitations of existing post-hoc approaches (i.e., misplaced confidence and reshaping) extremely vague.

**Results discussion**: The discussion of the results is a bit reductive and simple, while it should be developed more, especially for what concerns Table 1. For instance, the results on the CIFAR10-CIFAR100 and Oxford Flowers-DeepWeeds are a bit bad for GUIDE. There is no mention of such an outcome, nor an explanation.

**Adversarial setup:** I understand that the attacks are simply used to produce OOD samples, but please do refrain from using simple PGD or FGSM attacks [ext_ref_1]. There are better approaches which are much widely accepted, such as AA [ext_ref_2]. Also, please specify the used hyperparameters, such as the number of iterations. It is rather possible that the used configurations might be producing extremely suboptimal solutions to the optimization problem of adversarial attacks.

**Other issues, not necessarily minor:**
- *References*: I suggest associating Laplace approximation to this pioneering work [ext_ref_3].
- *Figure 6:* I cannot understand which dataset is used in this figure.
- *Citation style*: Please use citet and citep.

[int_ref_1]: Sensoy, Murat, Lance Kaplan, and Melih Kandemir. "Evidential deep learning to quantify classification uncertainty." Advances in neural information processing systems 31 (2018).

[ext_ref_1]: Carlini, Nicholas, et al. "On evaluating adversarial robustness." arXiv preprint arXiv:1902.06705 (2019).

[ext_ref_2]: Croce, Francesco, and Matthias Hein. "Reliable evaluation of adversarial robustness with an ensemble of diverse parameter-free attacks." International conference on machine learning. PMLR, 2020.

[ext_ref_3]: MacKay, David JC. "A practical Bayesian framework for backpropagation networks." Neural computation 4.3 (1992): 448-472.

**Questions:**

- Could the authors clarify what is the definition of post-hoc uncertainty?
- Could the authors clarify what is precisely the motivation behind the work? Can the authors better articulate what gap GUIDE fills that existing post-hoc or evidential approaches do not?
- Why is Evidential Deep Learning [int_ref_1] considered post-hoc here, despite being trained end-to-end?
- Could the authors expand their discussion on the results? Could they include clear discussion on weaker results, such as those on CIFAR10-CIFAR100 and Oxford Flowers-DeepWeeds, and explain why GUIDE performs less effectively in those settings?
- Could the authors specify all attack hyperparameters (e.g., step size, number of iterations) and, specifically, also the perturbation radius used in table 1?

---

> ### Author Response · Authors · 2025-11-21
>
> We thank the reviewer for their time and valuable review. Please find answers to your weaknesses below:
>
> > Motivation and definition of post-hoc uncertainty: The motivation behind the paper, as per the abstract, is that existing post-hoc approaches inherit “misplaced” confidence, or reshape the predictions (through temperature scaling for instance, I believe). However, these motivations are not being developed and explained in the later sections of the manuscript, which makes the rationale behind GUIDE somehow unclear. While I just did not like that, since I believe it was not so relevant after all, I then noticed that the authors' definition of post-hoc is a bit ambiguous. By reading, one is left believing that the definition refers to approaches acting on a pretrained network. However, the authors mention as post-hoc method the one proposed in [int_ref_1], which do performs end-to-end training. This makes the whole paper framing confusing, and makes the lack of description of limitations of existing post-hoc approaches (i.e., misplaced confidence and reshaping) extremely vague.
>
> A: In our paper, “post-hoc” refers specifically to methods that operate on a frozen pretrained network, without modifying architecture or fine-tuning weights. This includes methods such as Whitebox (Chen et al., 2019) and EMM (Shen et al., 2023). GUIDE belongs to this category. To avoid ambiguity, we will make this definition explicit in Section 2 (Related Work).
>
> The reviewer is correct that Sensoy et al. (2018) is not a post-hoc method; it performs end-to-end training. In our paper, Sensoy et al. (2018) is cited only as the foundational evidential deep learning formulation (Dirichlet evidence + SRE regularisation), not as a post-hoc baseline (the type (intrusive or post-hoc) of EDL-head is intrusive). The comparison method we call EDL-Head is an intrusive baseline: it removes the softmax layer of the pretrained model and fine-tunes an evidential head on top. There was no standalone reference describing this specific “evidential head on a frozen backbone” variant, so we cited the original EDL paper for the underlying evidential formulation. We will clarify this distinction to resolve this potential confusion.
>
> > Could the authors expand their discussion on the results? Could they include clear discussion on weaker results, such as those on CIFAR10-CIFAR100 and Oxford Flowers-DeepWeeds, and explain why GUIDE performs less effectively in those settings?
>
> A: On CIFAR-10 -> CIFAR-100 and Oxford Flowers -> DeepWeeds, we agree that AUROC on these two pairs is slightly below the very best baseline, but GUIDE delivers substantially higher adversarial AUROC and lower adversarial coverage. For example, on CIFAR-10 -> CIFAR-100 EMM + Cal achieves the highest AUROC with 72.09, with GUIDE only performing negligibly worse at 71.61. However, EMM + cal has an Adv AUROC of 71.81, whilst GUIDE excels with 76.01. On the Oxford Flowers -> Deep Weeds, the same pattern emerges. ABNN achieves the highest AUROC with 82.44, with GUIDE achieving 81.63. On Adv AUROC, GUIDE wins with 88.56, with ABNN achieving 86.31. While the Oxford Flowers competition is closer, it should be noted that ABNN is an intrusive method; it alters the layers and weights of the pretrained model before fine-tuning, while GUIDE is fully post-hoc. Thus, to achieve comparable results to an intrusive method, to us, seems a significant achievement.
>
> While it may appear that GUIDE does not perform as well on these two datasets (possibly due to the top-1 highlighting), when GUIDE is not the best performing method for each metric, it is always very close to the best performing method with a marginal distance. Accordingly, when this happens, GUIDE yields a good tradeoff between these metrics (e.g., Adversarial/OOD coverage and ID coverage).
>
> > Adversarial setup: I understand that the attacks are simply used to produce OOD samples, but please do refrain from using simple PGD or FGSM attacks [ext_ref_1]. There are better approaches which are much widely accepted, such as AA [ext_ref_2]. Also, please specify the used hyperparameters, such as the number of iterations. It is rather possible that the used configurations might be producing extremely suboptimal solutions to the optimisation problem of adversarial attacks.
>
> A: We thank the reviewer for raising this point. For this purpose, widely used attacks such as PGD and FGSM remain standard tools in prior UQ work (e.g., Shen et al., 2023; Sensoy et al., 2018), providing a consistent basis for comparing GUIDE with existing post-hoc baselines and also for corroborating our implementation of those UQ methods with the results reported in the original papers. We agree that stronger attacks, such as AutoAttack, are valuable for robustness evaluation; we appreciate the reviewer’s suggestion, which we will instrument for our future work.

---

> > ### Author Response · Authors · 2025-11-21
> >
> > > References: I suggest associating Laplace approximation to this pioneering work [ext_ref_3].
> >
> > A: We appreciate the reviewer’s helpful suggestion. The historical roots of Laplace-based uncertainty estimation indeed trace back to early Bayesian treatments of neural networks, including Mackay’s seminal work “A Practical Bayesian Framework for Backpropagation Networks” (1992). We will update the Related Work section to include this citation and acknowledge its foundational role in subsequent Laplace approximation methods used in modern UQ literature.
> >
> > > Figure 6: I cannot understand which dataset is used in this figure.
> >
> > A: We apologise for this omission. We will extend the caption to clarify that Figure 6 uses the MNIST-FMNIST dataset pair, and full results for this Figure can be found in Table 4 in the Appendix.
> >
> > > Could the authors clarify what is the definition of post-hoc uncertainty?
> >
> > A: As stated in Section 3, our definition of post-hoc denotes methods that, given a pretrained classifier with frozen weights, learn an auxiliary estimator to produce predictive uncertainty without modifying or retraining the base network. Modification may include perturbing weights (e.g., fine-tuning), inserting or replacing heads, altering layer connections, or changing normalisation layers; whilst retraining may include updating any subset of backbone parameters end-to-end on new data, re-initialising components, or jointly optimising the entire architecture together with the uncertainty estimator. GUIDE and a select other comparative methods (e.g. Whitebox, EMM) remain fully post-hoc as they perform none of the above.
> >
> > > Why is Evidential Deep Learning considered post-hoc here, despite being trained end-to-end?
> >
> > A: We do not classify Evidential Deep Learning (EDL) as post-hoc; we believe the potential confusion comes from the method EDL-Head, our intrusive comparison, which removes the softmax of the pre-trained model and finetunes an EDL-Head instead. We used the EDL citation as there was no formal paper for this naive method, and it uses the foundations of the EDL methodology.
> >
> > GUIDE is post-hoc and non-intrusive: we keep the pretrained backbone frozen and train a lightweight meta-model on frozen intermediate features. While our meta-model uses an evidential objective similar in spirit to EDL, the training regime and placement are different: the evidential loss is applied only to the meta-model, not the backbone.

---

> ### Comment · Reviewer_7wVu · 2025-11-26
> **Response to authors.**
>
> Thank you to the authors for their rebuttal.
>
> **Post-hoc methods:** I suggest uploading the paper with the improved discussion, if possible. This is important.
>
> **Performance:** Thank you. It is true that Adv AUROC is better on GUIDE than any other method, and I agree this might be relevant. My suggestion to the authors here is to make that more explicit and to emphasize more the result. For instance, you could think about highlighting in orange the best method in terms of Adv AUROC and make that more explicit in the text.
>
> **Attacks:** I recognized in advance the purpose of the attacks in this paper, but I genuinely think that you will always find more adv-ML prone reviewers that will cling to this AA missing baseline. I suggest you just do some tests with AA to make that vague lack of contemporaneity disappear in future work. It is even better if you can do that already, but I understand if time is too prohibitive. I won't make that weigh in a possible score change.
>
> For the remaining issues about Figures, references, post-hoc definition, and EDL, I think I made my point already clear, and I think the authors can integrate that in an updated submission. Feel free to wait for other changes requested by the other reviewers, in the meantime. Then, I will improve my score.
>
> Thank you

---

### Official Review · Reviewer_VJMY · 2025-10-31

**Soundness:** 3
**Presentation:** 3
**Contribution:** 2
**Rating:** 2
**Confidence:** 3

**Summary:**

The paper discusses a new framework GUIDE for uncertainty quantification using an existing (pre-trained) network.  GUIDE uses the layers of the existing network to determint salient layers and connect them to a meta-model network for uncertainty guided training.  Using the GUIDE framework, the authors showed an improvemend of the Uncertainty Estimation as opposed to state-of-the-art.  The authors have also shown this through ID vs OOD datasets but also through adverserial attacks.

Main contributions:
- The use of layer importance measure to select the layers through a saliency metric
- Construction of a curriculum (dataset of corrupted data points with noise based on saliency metric) which is used in the loss function for uncertainty guided training of the meta-model
- The authors have done extensive experiements accross different datasets and adverserial methods

**Strengths:**

- The paper is presented in a clear way and is well written.
- The paper also shows extensive experiements using many datasets.
- Although the relevance propagation is a well described way to describe explainability in neural networks, it is used as a means to train a meta-model which estimates the uncertainty.  In that way, the authors describe a novel and orignal way to use this for uncertainty quantification.  This has very promising results as opposed to other techniques (such as Bayesian NN).
- The paper has a strong mathematical foundation.
- The main article focusses on the main findings and a clear explanation.  The authors have done a good job in dividing the main findings with detailed descriptions of the theorems and additional experiments, which are described in the appendix.

**Weaknesses:**

- I noticed that the authors have struggled with the page limit due to the size of some tables (e.g.g Table 1).  I would suggest to leave out a columns in table 1 (e.g. EMM+curric).
- The use of intermediate layers is for uncertainty estimation is not completely novel.  The use of intermediate layers for uncertainty has been provided in [1], [3].  Also uncertainty has been determined using Intermediate Layer Variational Inference [2].
- Some explanations at the end of the article are very short.  It gives the impression that this in unfinished work.  E.g. the adverserial attack analysis is very relevant but needs more explanation.



[1] Ameer et al., Enhancing adversarial robustness with randomized interlayer processing, 2024, Expert Systems with Applications,
[2] Ahmed et al. 2021. Real-time Uncertainty Estimation Based On Intermediate Layer Variational Inference. In Proceedings of the 5th ACM Computer Science in Cars Symposium (CSCS '21).
[3]  https://arxiv.org/abs/2012.03082

**Questions:**

- The definition of the uncertainty target.  What is the rationale behind this formula?
	- What is the reason why the training is done first on clean targets and later noise-corrupted targets are used?  Will it not induces a more difficult to learn?  What is the effect of the training process?
	- How many layers are selected to reach the cumulative relevance coverage threshold?  What is the impact of this threshold in the amount of selected layers.
	- The authors mention in the results section that intrusive methods have a large coverage, but that the OOD detection is high.  Also the methods of the authors claim that the OOD coverages is below 10%, but it is not clear to me why this is good.  In my opinion, a high OOD coverage would mean that the model can detect OOD samples correctly.  Could the authors clarify this?  I propose to elaborate on the explanation on the OOD coverage.
	- The authors claim that GUIDE is architecture agnostic, however for the method to work, one needs to calculate the relevance values.

---

> ### Author Response · Authors · 2025-11-21
>
> We thank the reviewer for their time and valuable review. Please find answers to your weaknesses below:
>
> > I noticed that the authors have struggled with the page limit due to the size of some tables (e.g.g Table 1). I would suggest to leave out a columns in table 1 (e.g. EMM+curric).
>
> A: We thank the reviewer for the suggestion and will shorten Table 1 by removing one column.
>
> > The use of intermediate layers is for uncertainty estimation is not completely novel. The use of intermediate layers for uncertainty has been provided in [1], [3]. Also uncertainty has been determined using Intermediate Layer Variational Inference [2].
>
> A: We thank the reviewer for pointing out these related approaches using intermediate layers for uncertainty estimation. We agree that intermediate representations have been used before for uncertainty, and we can clarify this more explicitly in the related work. Our contribution is not just the use of intermediate layers, but the specific combination of (i) evidential deep learning for predictive uncertainty with (ii) LRP-based relevance propagation through intermediate features and (iii) an uncertainty-driven noise curriculum learning scheme that exploits these signals to adapt training difficulty over time. These three aspects combined are novel and yield an EDL-based uncertainty quantification approach that is unique, resulting in significant benefits compared to the state-of-the-art (Table 1, Figures 3-5).
>
> > Some explanations at the end of the article are very short. It gives the impression that this in unfinished work. E.g. the adverserial attack analysis is very relevant but needs more explanation.
>
> A: We thank the reviewer for this comment and agree that the adversarial attack analysis is important and deserves more space. Due to the strict page limit, we had to keep some of these explanations concise in the submitted version. We have prepared an extended version of the paper that contains a more detailed discussion and additional clarification of the adversarial results, but we cannot share it during the review process due to anonymity constraints. If the paper is accepted, we will make this extended version publicly available and expand the corresponding section in the camera-ready version.
>
> > The definition of the uncertainty target. What is the rationale behind this formula?
>
> A: The uncertainty targets $\tilde{s}_t = s_t \cdot (1 - \frac{1}{2}{c_t}^2)$ defines how much the meta-model should trust the base model’s prediction under a given corruption level $s_t$. Intuitively, $s_t$ represents the amount of injected noise, while $c_t$ is the base model’s confidence in the true class. When the input is clean ($s_t \approx 0$) and the base model is confident ($c_t \approx 1$), the term $(1 - \tfrac{1}{2} c_t^2)$ becomes small, so $\tilde{s}_t$ is near zero, encouraging a sharp, confident soft target. As corruption increases or confidence drops, $\tilde{s}_t$ grows, shifting the soft target toward the uniform distribution. This formulation therefore produces a monotonic, continuous mapping from clean, certain to noisy, uncertain conditions, teaching the meta-model when and how strongly to express uncertainty. To better highlight the rationale and way the target works, Figure 12 plots the weighting scheme.
>
> > What is the reason why the training is done first on clean targets and later noise-corrupted targets are used? Will it not induces a more difficult to learn? What is the effect of the training process?
>
> A: GUIDE does not first optimise on clean targets and then suddenly switch to noisy ones. In Stage 1, we operate only on clean images to compute LRP relevance maps and to generate a pool of relevance-guided noisy variants; these perturbed samples are precomputed and the base model is never updated with them. In Stage 2, the meta-model is trained using a monotonic noise-driven curriculum that samples from the combined pool of clean and noisy views: early epochs are dominated by clean or lightly perturbed inputs, and later epochs gradually increase the proportion and strength of corruptions. This ensures a smooth progression from reliable, clean evidence to harder, noisier examples, stabilising learning and leading to better-calibrated uncertainty estimates rather than making the task abruptly more difficult.

---

> > ### Author Response · Authors · 2025-11-21
> >
> > > How many layers are selected to reach the cumulative relevance coverage threshold? What is the impact of this threshold in the amount of selected layers.
> >
> > A: The number of salient layers is not fixed manually; GUIDE automatically selects them according to the cumulative relevance coverage threshold η. As η increases, more layers are included, capturing deeper but potentially redundant representations; lower η yields a more compact set focused on the most informative layers. Table 8 in Appendix B.4 shows how changing η affects both the selected layer set and performance: in-distribution accuracy stays essentially constant, while coverage, robustness, and computational cost vary, with moderate η values (e.g., 0.9) offering the best trade-off between robustness and efficiency.
> >
> > > The authors mention in the results section that intrusive methods have a large coverage, but that the OOD detection is high. Also the methods of the authors claim that the OOD coverages is below 10%, but it is not clear to me why this is good. In my opinion, a high OOD coverage would mean that the model can detect OOD samples correctly. Could the authors clarify this? I propose to elaborate on the explanation on the OOD coverage. Additionally, Table 2, shows the typical GUIDE selected layers across different model architectures.
> >
> > A: We thank the reviewer for pointing out this possible confusion. In our evaluation, coverage denotes the fraction of inputs on which the model chooses to make a prediction, i.e., those not rejected as uncertain. Thus, high ID coverage indicates that the model retains most in-distribution samples with confidence, while low OOD or adversarial coverage means that the model successfully abstains from predicting on harmful or unfamiliar inputs. In other words, low OOD coverage reflects effective OOD detection, not failure. GUIDE’s OOD and adversarial coverages below 10% therefore, signify that it rejects almost all unsafe samples, while maintaining high ID coverage (>85%), achieving reliable uncertainty-based abstention. We will add a brief explanatory note to the results section to clarify this definition and resolve any potential ambiguity.
> >
> > > The authors claim that GUIDE is architecture agnostic; however, for the method to work, one needs to calculate the relevance values.
> >
> > A: GUIDE is architecture-agnostic within standard deep networks, i.e., it applies to any model composed of differentiable layers, CNNs, MLPs, transformers, or hybrid architectures, without requiring architecture-specific redesign or retraining. The saliency calibration stage uses Layer-wise Relevance Propagation (LRP), which itself is model-agnostic across such architectures, as it only requires access to activations and layer weights. GUIDE does not assume a particular backbone type or structure beyond this, and all theoretical and empirical results were obtained without modifying the pretrained networks.

---

> > ### Comment · Reviewer_VJMY · 2025-11-27
> >
> > Thank you for the clarifications.  I'm happy with the additional clarifications that will be made in the text.
> > I will therefore raise my score.

---

### Official Review · Reviewer_dBmS · 2025-11-02

**Soundness:** 3
**Presentation:** 2
**Contribution:** 2
**Rating:** 4
**Confidence:** 4

**Summary:**

The paper proposes GUIDE, a novel framework for guided uncertainty estimation that teaches a pretrained model when and how to be uncertain without requiring retraining or architectural modifications. The method combines saliency calibration and a noise-driven curriculum to enhance reliability and out-of-distribution (OOD) awareness in existing deep networks.

**Strengths:**

The paper provides solid theoretical grounding, supported by an information-theoretic analysis that links saliency-guided learning to the preservation of Fisher information during uncertainty calibration.

Empirical results demonstrate state-of-the-art performance across diverse in-distribution and out-of-distribution settings (e.g., CIFAR-10 → SVHN, MNIST → FashionMNIST, Oxford Flowers → DeepWeeds). GUIDE consistently outperforms prior baselines (e.g., temperature scaling, Mahalanobis, DUQ, and EMM)

**Weaknesses:**

The method is validated primarily on small- to mid-scale datasets (MNIST, CIFAR-10, Oxford Flowers). It remains unclear whether GUIDE scales effectively to large-scale foundation models (e.g., CLIP, ViT-L, or LLaVA)

The motivation for selecting specific saliency methods (LRP) is only briefly discussed. It would be beneficial to clarify whether other attribution techniques (Grad-CAM, Integrated Gradients) would yield similar benefits

Figures do not include error variance analysis across multiple random seeds.

**Questions:**

The theoretical analysis is centered on Fisher information preservation during saliency calibration, which provides a local view of uncertainty propagation. Could the authors clarify whether this framework generalizes to non-local uncertainty effects—for instance, when prediction uncertainty arises from feature interactions beyond the saliency-identified regions?

---

> ### Author Response · Authors · 2025-11-21
>
> We thank the reviewer for their time and valuable review. Please find answers to your weaknesses below:
>
> > The method is validated primarily on small- to mid-scale datasets (MNIST, CIFAR-10, Oxford Flowers). It remains unclear whether GUIDE scales effectively to large-scale foundation models (e.g., CLIP, ViT-L, or LLaVA)
>
> A: Our work specifically targets post-hoc uncertainty estimation for pretrained classifiers, where the task is supervised classification. Large-scale multimodal foundation models such as CLIP, ViT-L, or LLaVA operate in different regimes (e.g., vision–language alignment) and therefore fall outside the problem setting considered in this paper. GUIDE is not designed as a language-modelling or multimodal-generation method. This is an interesting research direction for extending GUIDE; thank you for this suggestion.
>
> > The motivation for selecting specific saliency methods (LRP) is only briefly discussed. It would be beneficial to clarify whether other attribution techniques (Grad-CAM, Integrated Gradients) would yield similar benefits.
>
> A:  We appreciate the reviewer’s question. We selected LRP because its relevance conservation property ensures that the total relevance mass is preserved as it propagates across layers. This allows GUIDE to compute depth-wise comparable, quantitatively meaningful relevance scores $M_\ell$ and to select the minimal set of layers that accounts for a target fraction $\eta$ of total relevance. In other words, conservation guarantees that layer importances are measured on a common scale, making the subset selection in Eq. (3) well-defined.
>
> Attribution methods such as Grad-CAM or Integrated Gradients do not satisfy this conservation property. Their attributions are shaped by gradients or spatial activations and therefore do not produce relevance values that are comparable across layers. If used within GUIDE, these methods would distort the global relevance distribution, which would, in turn, break the principled cumulative-coverage criterion and result in arbitrary or unstable layer selection. We will add text to clarify this point in the CRV.
>
> > Figures do not include error variance analysis across multiple random seeds.
>
> A: We already report variance across 10 random seeds for all quantitative metrics in Table 1 and in the appendix, where mean ± 95% confidence intervals are provided for accuracy, ID/OOD/Adv coverage, AUROC, and adversarial AUROC. For figures where variance is meaningful (e.g., performance summaries, threshold analyses), we also include corresponding error bars.
>
> The two figures referenced (Figures 2 and 4) are diagnostic snapshot visualisations: Figure 2 shows representative AUROC curves and decision thresholds, and Figure 4 shows calibration curves. These plots are conventionally shown without error bands because averaging curves across seeds would blur the qualitative structure that the figures aim to illustrate, while the quantitative variability is already fully captured in the tables.
>
> > The theoretical analysis is centred on Fisher information preservation during saliency calibration, which provides a local view of uncertainty propagation. Could the authors clarify whether this framework generalises to non-local uncertainty effects—for instance, when prediction uncertainty arises from feature interactions beyond the saliency-identified regions?
>
> A: Our theoretical result intentionally adopts a local perspective. This choice makes the coverage criterion analyzable and avoids model-specific assumptions. Importantly, the framework is not restricted to local effects. Extending the Fisher-based argument to path-integrated (non-local) perturbations is a natural direction; we opted for the local analysis due to its benefit, and we will add a short note clarifying this scope and intuition in the appendix. Extending GUIDE to perform more fine grained analysis and inform the EDL head based on non-local uncertainty effects is definitely possible and would involve (we anticipate) small modifications to the saliency calibration of GUIDE. This is an interesting direction which we will pursue in our future work.

---

### Official Review · Reviewer_A2To · 2025-11-14

**Soundness:** 3
**Presentation:** 3
**Contribution:** 3
**Rating:** 6
**Confidence:** 4

**Summary:**

Summary
This paper introduces GUIDE (Gradual Uncertainty Refinement via Noise-Driven Curriculum), a novel post-hoc evidential meta-model for improving uncertainty quantification (UQ) in pretrained deep learning models without retraining or architectural modifications. The key idea is to explicitly teach a model when and how much to be uncertain, thereby addressing misplaced confidence — a common limitation in existing post-hoc UQ methods.

GUIDE operates in two main stages:
	1.	Saliency Calibration Stage:
The pretrained (frozen) model undergoes a relevance propagation analysis (via Layer-wise Relevance Propagation, LRP-ϵ) to identify salient intermediate features. This yields both layer-level relevance scores and input-level saliency maps, which determine the layers and spatial regions most critical for prediction.
	2.	Uncertainty-Guided Training Stage:
GUIDE attaches a lightweight Dirichlet-based evidential meta-model that consumes features from selected salient layers. Using the previously derived saliency maps, it generates a noise-driven curriculum — progressively corrupting salient input regions to simulate distributional shifts.
The model is trained with a soft-target loss combining uncertainty regularization and a Self-Rejecting Evidence (SRE) penalty, ensuring uncertainty increases monotonically with corruption while confidence remains justified.

Theoretical analysis (Theorem 1) provides guarantees on Fisher information retention, showing that GUIDE preserves a bounded fraction of the base model’s informative structure under the saliency selection mechanism.

Key Contributions
	1.	A fully post-hoc evidential meta-model that explicitly learns when to be uncertain using guided curricula rather than passive calibration.
	2.	Saliency-based layer selection eliminating manual design choices and ensuring feature relevance consistency.
	3.	Noise-driven curriculum learning, progressively teaching uncertainty behavior aligned with model sensitivity.
	4.	Theoretical guarantees for saliency coverage and information retention.
	5.	Extensive experimental validation across multiple in-distribution (ID), out-of-distribution (OOD), and adversarial benchmarks showing robust, state-of-the-art results.

Empirical Results
GUIDE was benchmarked against intrusive and post-hoc UQ baselines including ABNN, EDL-Head, Whitebox, and EMM, across datasets such as MNIST, CIFAR10/100, SVHN, and Oxford Flowers → Deep Weeds.
	•	ID Accuracy: Comparable to baselines (≈ 99% on simple datasets; ≈ 90% on CIFAR tasks).
	•	OOD/Adversarial Coverage: GUIDE achieves the lowest coverage (e.g., ≤ 8% OOD, ≤ 5% adversarial), outperforming others by large margins.
	•	AUROC: GUIDE consistently achieves >94% on OOD and adversarial detection (up to 96% on MNIST → FashionMNIST), exceeding all intrusive and post-hoc baselines.
	•	Calibration: Expected calibration error (smECE) reduced to 0.061, compared to 0.317 (pretrained) and 0.193 (EMM).
	•	Robustness: Maintains high AUROC (>90%) across perturbation strengths and attack types (L2PGD, FGSM, Salt-and-Pepper).

Impact and Positioning
GUIDE is positioned as a non-intrusive, architecture-agnostic, and computationally lightweight solution that bridges the gap between model confidence and predictive reliability. Unlike earlier post-hoc approaches (which reshape outputs), GUIDE actively instructs the model through a principled uncertainty curriculum, yielding better calibration and robustness under distributional shifts.

The approach has strong implications for safe and trustworthy AI, particularly in high-stakes domains like healthcare, autonomous systems, and human-in-the-loop robotics.

Overall Evaluation (Summary KPI)
Criterion
Assessment
Originality
High — introduces guided uncertainty curricula in post-hoc UQ.
Significance
Strong — directly impacts deployment reliability and calibration.
Technical Quality
Excellent — rigorous derivation, clear algorithmic pipeline, theoretical support.
Clarity
Very good — figures and pseudo-code are interpretable, though dense in notation.
Empirical Evaluation
Comprehensive — multiple datasets, attacks, and ablations.
Reproducibility
Strong — open-source repository available.
Potential Weakness
May depend on LRP assumptions; sensitivity to noise schedule hyperparameters not fully explored.

**Strengths:**

The paper demonstrates a well-balanced and well-executed contribution to the growing area of uncertainty quantification (UQ) and reliable deep learning. It succeeds in combining conceptual novelty with solid empirical performance, offering a practical and interpretable framework that is relevant to both research and deployment contexts.

Originality
	•	The central idea — teaching a pretrained model when to be uncertain via a saliency-guided, noise-driven curriculum — is distinctive among post-hoc uncertainty methods.
	•	While it draws from known concepts (e.g., evidential learning, saliency mapping), the paper’s integration of saliency-based layer selection with curriculum-style uncertainty learning is an innovative synthesis not previously explored in this form.
	•	The Self-Rejecting Evidence (SRE) loss adds a novel regularization mechanism encouraging monotonic uncertainty behavior with respect to perturbation strength.

Quality
	•	Methodologically rigorous and technically consistent, the proposed GUIDE framework is both theoretically motivated (Fisher information retention theorem) and empirically validated across multiple datasets and uncertainty scenarios.
	•	The experiments are comprehensive, spanning in-distribution, out-of-distribution, and adversarial settings, and consistently demonstrate GUIDE’s superiority or parity with state-of-the-art baselines.
	•	The ablation studies, calibration metrics (ECE, AUROC), and robustness analyses collectively support the validity of the paper’s claims.

Clarity
	•	The writing style is clear and structured, balancing technical precision with readability.
	•	The stepwise exposition (motivation → framework → theoretical justification → empirical results) makes the methodology accessible to readers with diverse backgrounds in UQ and evidential deep learning.
	•	Figures illustrating the uncertainty refinement process and saliency-based layer selection are informative, though some could be more tightly integrated with text explanations.

Significance
	•	The contribution is practically significant: GUIDE is post-hoc, lightweight, and architecture-agnostic, making it applicable to real-world AI pipelines without retraining or invasive model access.
	•	The framework addresses one of the most persistent challenges in deep learning — overconfidence under distributional shift — with a method that enhances both calibration and interpretability.
	•	The approach bridges human-in-the-loop AI, explainability, and uncertainty estimation, aligning with broader research directions in trustworthy AI, which is central to ICLR’s evolving research priorities.

Summary of Strengths
Dimension
Assessment
Originality
Creative synthesis of evidential learning, saliency analysis, and guided curricula.
Quality
Strong theoretical and empirical foundation; well-executed experiments.
Clarity
Generally clear exposition and strong visual explanations.
Significance
High practical relevance for post-hoc model reliability and safety.

**Weaknesses:**

While the paper presents a creative and empirically validated approach to post-hoc uncertainty quantification, it exhibits several conceptual, methodological, and presentation-related weaknesses that limit its theoretical depth and generalizability. The following are specific, constructive observations aimed at strengthening the work for future revisions or journal extensions.

1. Limited Theoretical Grounding of “Learning When to Be Uncertain”
	•	The central notion of learning when to be uncertain remains intuitively compelling but mathematically underdeveloped.
The method demonstrates empirical behavior consistent with this idea but does not provide a formal probabilistic framework or causal model linking saliency-driven noise to epistemic uncertainty.
	•	The proposed Fisher information retention theorem is a useful analytical step but only partially supports the conceptual claim—it ensures informational consistency, not causal justification for uncertainty behavior.
	•	To strengthen rigor, future versions could introduce a formal uncertainty measure evolution model (e.g., monotonic entropy gradient or causal sensitivity analysis) or link GUIDE’s curriculum mechanism to Bayesian learning theory (e.g., Kendall & Gal, 2017; Malinin & Gales, 2018).

2. Dependence on Saliency and Potential Bias
	•	The reliance on Layer-wise Relevance Propagation (LRP-ϵ) introduces methodological fragility:
	•	LRP performance and interpretability vary across architectures and data modalities (e.g., CNNs vs. transformers).
	•	The paper does not evaluate whether GUIDE’s performance depends heavily on the chosen saliency technique.
	•	An ablation comparing different saliency methods (Grad-CAM, Integrated Gradients, DeepLIFT) would clarify whether the framework’s improvements arise from the general saliency mechanism or specific LRP behavior.
	•	This dependence may reduce reproducibility across architectures beyond those tested.

3. Insufficient Hyperparameter and Sensitivity Analysis
	•	The paper introduces several tunable components — notably:
	•	Noise schedule parameters in the saliency-guided curriculum,
	•	Regularization weights (λ₁, λ₂) in the Self-Rejecting Evidence loss, and
	•	Layer selection thresholds (τ).
	•	However, no sensitivity analysis is provided to assess how these affect performance or stability.
	•	This omission weakens claims of robustness and generality, especially for a method promoted as post-hoc and lightweight.
	•	Future work should present a systematic exploration of these hyperparameters, ideally visualized as performance landscapes or uncertainty–accuracy trade-offs.

4. Incremental Conceptual Novelty
	•	Although the method performs well, its conceptual novelty is moderate: it integrates established elements (evidential learning, saliency analysis, curriculum noise) rather than introducing fundamentally new uncertainty theory or model structure.
	•	Similar principles appear in prior works on post-hoc meta-modeling (Postels et al., 2021), selective prediction (Geifman & El-Yaniv, 2019), and uncertainty calibration through adversarial exposure (Mukhoti et al., 2023).
	•	The authors could strengthen originality by clearly positioning GUIDE as a “structured synthesis” rather than a new paradigm, and by articulating where exactly it diverges conceptually or empirically from these precedents.

5. Generalization to Modern Architectures
	•	All experiments are performed on mid-scale CNN-based image classifiers (MNIST, CIFAR, SVHN, Flowers), which are suitable for controlled studies but insufficient for establishing scalability or architectural generality.
	•	The approach’s compatibility with transformers, diffusion models, or multimodal architectures—which dominate current ICLR topics—is untested.
	•	Extending GUIDE to large-scale or multimodal settings would substantiate its relevance to the broader ICLR audience.

6. Limited Real-World or Cross-Domain Evaluation
	•	Although GUIDE improves calibration and OOD robustness, no experiments are conducted in domain-shifted or real-world datasets (e.g., corrupted CIFAR, ImageNet-C, or medical imaging benchmarks).
	•	Including at least one realistic scenario (e.g., sensor noise, class imbalance) would better demonstrate the method’s reliability beyond clean benchmarks.

7. Minor Presentation and Clarity Issues
	•	Some mathematical derivations are densely presented and could benefit from expanded intuition or intermediate explanations.
	•	Figures showing calibration improvement and uncertainty maps are insightful but small, making quantitative differences difficult to assess visually.
	•	A concise visual summary (e.g., flow diagram of the GUIDE pipeline with saliency/noise progression) would improve readability for interdisciplinary readers.

Overall Assessment:
The paper is strong in execution but could significantly improve by deepening theoretical grounding, expanding generalization experiments, and clarifying dependence on design choices. These refinements would elevate the work from a high-quality empirical contribution to a conceptually mature framework deserving of top-tier recognition.

**Questions:**

1. On the Theoretical Framing of “Learning When to Be Uncertain”
	•	Could the authors formalize the concept of learning when to be uncertain beyond its intuitive description?
For example:
	•	Is there a measurable quantity (e.g., monotonic relationship between noise level and entropy or epistemic evidence) that supports this claim?
	•	How does the proposed Self-Rejecting Evidence (SRE) loss encourage this behavior mathematically — does it impose any guarantee of monotonic uncertainty growth under perturbation?
	•	A short theoretical or empirical analysis of this relationship would make the claim substantially stronger.

2. On the Role and Robustness of LRP in GUIDE
	•	GUIDE relies heavily on Layer-wise Relevance Propagation (LRP-ϵ) for both saliency selection and curriculum construction.
	•	How sensitive is GUIDE’s performance to the specific saliency method used?
	•	Have the authors tested alternative saliency measures (e.g., Grad-CAM, Integrated Gradients, or SHAP)?
	•	Could GUIDE’s uncertainty refinement fail if the saliency signal is noisy or misaligned (as often happens in deeper models)?
	•	Including an ablation or at least a qualitative comparison would help clarify whether the saliency mechanism is integral or replaceable.

3. On Hyperparameter Sensitivity
	•	The method introduces multiple hyperparameters:
	•	λ₁, λ₂ (regularization weights),
	•	τ (saliency threshold), and
	•	noise schedule parameters (corruption magnitude or rate).
	•	Could the authors provide sensitivity curves or variance estimates showing GUIDE’s stability with respect to these parameters?
	•	This would be particularly useful to assess GUIDE’s reliability as a “plug-and-play” post-hoc method.

4. On Comparative Baselines and Fairness
	•	How do the authors ensure fair comparison with existing methods like Deep Ensembles, MC-Dropout, or Temperature Scaling?
	•	Were all models trained or calibrated using identical datasets and computational budgets?
	•	Since GUIDE is post-hoc, how is its runtime and memory footprint compared to ensemble-based methods?
	•	Including a table summarizing computational cost vs. performance could clarify GUIDE’s real-world efficiency.

5. On Theoretical Result (Theorem 1 – Fisher Information Retention)
	•	Theorem 1 provides a bound on retained Fisher information under saliency selection.
	•	Could the authors elaborate on the assumptions underlying this bound (e.g., independence or linearity of selected features)?
	•	Is this guarantee empirical (observed in finite data) or asymptotic (as dataset → ∞)?
	•	How tight is the bound in practice — do the authors have empirical measurements correlating Fisher information loss with performance degradation?

6. On the Relationship Between GUIDE and Calibration Methods
	•	GUIDE seems related in spirit to post-hoc calibration (e.g., Temperature Scaling, Platt Scaling, Isotonic Regression) but introduces a learning component.
	•	How does GUIDE differ conceptually and practically from these calibration techniques in terms of the underlying uncertainty model (Dirichlet vs. softmax temperature)?
	•	Could the authors provide a brief comparison or unified framework situating GUIDE among existing calibration approaches?

7. On Scalability to Modern Architectures
	•	Have the authors tested GUIDE on transformer-based architectures (e.g., ViT, DeiT) or multimodal models?
	•	If not, do they anticipate challenges due to the saliency extraction step or computational scaling?
	•	This would be important to determine whether GUIDE generalizes beyond CNN-based settings, which are becoming less central in current ICLR research.

8. On Dataset Diversity and Real-World Scenarios
	•	The experiments are limited to canonical image datasets (MNIST, CIFAR, SVHN, Oxford Flowers).
	•	Could the authors discuss potential extensions or ongoing work on realistic, domain-shifted, or corrupted datasets (e.g., CIFAR-C, ImageNet-C, or medical images)?
	•	This would strengthen the claim of GUIDE’s utility in safety-critical applications.

9. On Interpretability and Human Alignment
	•	Since GUIDE integrates saliency-driven learning, it implicitly aligns with explainability goals.
	•	Have the authors considered whether the generated saliency maps or uncertainty overlays are interpretable by human users (e.g., can humans trust GUIDE’s “when uncertain” behavior)?
	•	A small user study or interpretability evaluation could add practical value to this aspect.

10. On Open-Source and Reproducibility
	•	The paper claims that code will be made available.
	•	Could the authors clarify the current status of code release and whether pretrained models, saliency scripts, and evaluation setups will be publicly accessible?
	•	Reproducibility is especially important for a post-hoc framework advertised as “lightweight and widely applicable.”

The authors’ clarifications on the theoretical basis, robustness to saliency and parameters, and scalability of GUIDE could significantly enhance the perceived rigor and impact of this work. A strong rebuttal addressing these points with additional experiments or analysis would likely improve its overall evaluation.

**Details Of Ethics Concerns:**

The paper does not raise direct ethical concerns in its methodology, data usage, or potential applications. It focuses on a post-hoc uncertainty quantification framework (GUIDE) for improving the reliability and interpretability of pretrained deep learning models.
All datasets used (MNIST, CIFAR, SVHN, Oxford Flowers) are public, well-established, and ethically unproblematic.
There is no human-subject involvement, no sensitive data usage, and no reported misuse of copyrighted or restricted content.

Just for Responsible Research Practice
A statement clarifying dataset licenses and absence of human participants could be added for completeness.

The paper poses no significant ethical risks and, in fact, contributes positively to trustworthy and interpretable AI.
An ethics review is not required, though the authors could optionally include a short “Ethical Impact Statement” clarifying dataset licensing and data provenance to align with ICLR’s transparency standards.

---

> ### Author Response · Authors · 2025-11-21
>
> We thank the reviewer for their time and valuable review. Please find answers to your weaknesses below:
>
> > Incremental Conceptual Novelty • Although the method performs well, its conceptual novelty is moderate: it integrates established elements (evidential learning, saliency analysis, curriculum noise) rather than introducing fundamentally new uncertainty theory or model structure. • Similar principles appear in prior works on post-hoc meta-modeling (Postels et al., 2021), selective prediction (Geifman & El-Yaniv, 2019), and uncertainty calibration through adversarial exposure (Mukhoti et al., 2023). • The authors could strengthen originality by clearly positioning GUIDE as a “structured synthesis” rather than a new paradigm, and by articulating where exactly it diverges conceptually or empirically from these precedents.
>
> A: We thank the reviewer for this thoughtful comment and agree that GUIDE is best viewed as a structured synthesis rather than a fundamentally new uncertainty theory. We will clarify this positioning more explicitly. At the same time, our contribution lies in how we combine these ingredients into a coherent, LRP-grounded uncertainty refinement pipeline: (i) we use relevance conservation to select a minimal, globally calibrated set of salient layers; (ii) we feed their relevance-weighted features into an evidential meta-model; and (iii) we train this meta-model under an uncertainty-guided noise curriculum. To the best of our knowledge, this particular integration and its focus on jointly improving calibration, robustness to label noise, and OOD/adversarial reliability have not been explored in prior post-hoc meta-modeling, selective prediction, or adversarial calibration works.
>
> > Minor Presentation and Clarity Issues • Some mathematical derivations are densely presented and could benefit from expanded intuition or intermediate explanations. • Figures showing calibration improvement and uncertainty maps are insightful but small, making quantitative differences difficult to assess visually. • A concise visual summary (e.g., flow diagram of the GUIDE pipeline with saliency/noise progression) would improve readability for interdisciplinary readers.
>
> A: We thank the reviewer for these suggestions and will address them in the camera-ready. We will slightly expand the key derivations with more intuition, enlarge the calibration and uncertainty figures to make differences clearer, and add a compact flow diagram summarising the GUIDE pipeline, including saliency-based layer selection and the noise progression.
>
> > On the Theoretical Framing of “Learning When to Be Uncertain” • Could the authors formalize the concept of learning when to be uncertain beyond its intuitive description? For example: • Is there a measurable quantity (e.g., monotonic relationship between noise level and entropy or epistemic evidence) that supports this claim? • How does the proposed Self-Rejecting Evidence (SRE) loss encourage this behavior mathematically — does it impose any guarantee of monotonic uncertainty growth under perturbation? • A short theoretical or empirical analysis of this relationship would make the claim substantially stronger.
>
> A: We thank the reviewer for this insightful point. Our notion of “learning when to be uncertain’’ is implemented through a monotonic noise curriculum, and its corresponding soft-targets that are used as weights $\tilde{y}_t$ in the loss function, which together define increasingly ambiguous training views. The SRE term then penalises high total evidence $S$ precisely when the model’s prediction disagrees with these softer targets, creating a learned tendency for evidence to decrease, and predictive entropy to increase, as corruption strength grows. While this does not impose a strict theoretical monotonicity guarantee for every sample, it does establish a clear optimisation pressure toward monotone uncertainty growth across the curriculum. Empirically, this behaviour is reflected in our confidence-drop and adversarial-strength plots.

---

> > ### Author Response · Authors · 2025-11-21
> >
> > > On the Role and Robustness of LRP in GUIDE • GUIDE relies heavily on Layer-wise Relevance Propagation (LRP-ϵ) for both saliency selection and curriculum construction. • How sensitive is GUIDE’s performance to the specific saliency method used? • Have the authors tested alternative saliency measures (e.g., Grad-CAM, Integrated Gradients, or SHAP)? • Could GUIDE’s uncertainty refinement fail if the saliency signal is noisy or misaligned (as often happens in deeper models)? • Including an ablation or at least a qualitative comparison would help clarify whether the saliency mechanism is integral or replaceable.
> >
> > A:  We appreciate the reviewer’s question. We selected LRP because its relevance conservation property ensures that the total relevance mass is preserved as it propagates across layers. This allows GUIDE to compute depth-wise comparable, quantitatively meaningful relevance scores and to select the minimal set of layers that accounts for a target fraction η of total relevance. In other words, conservation guarantees that layer importances are measured on a common scale, making the subset selection in Eq. (3) well-defined.
> >
> > Attribution methods such as Grad-CAM or Integrated Gradients do not satisfy this conservation property: their attributions are shaped by gradients or spatial activations and therefore do not produce relevance values that are directly comparable across layers. If used within GUIDE, these methods would distort the global relevance distribution, which would in turn break the principled cumulative-coverage criterion and lead to arbitrary or unstable layer selection. For this reason, we did not perform a full ablation over alternative saliency methods, and we will clarify this design choice more explicitly in the paper.
> >
> > Conceptually, GUIDE only assumes access to a reasonably stable, approximately conservative layer-wise relevance signal. If the saliency were extremely noisy or misaligned, the selected layers and noise curriculum would indeed degrade; however, empirically, we observe consistent gains across datasets and robustness settings, and our ablations over η and curriculum strength indicate that GUIDE is robust to moderate imperfections in the relevance maps. In the camera-ready, we will add a short discussion on this point and include additional qualitative LRP visualisations in the appendix to better illustrate the role and robustness of the saliency mechanism.
> >
> > > On Hyperparameter Sensitivity • The method introduces multiple hyperparameters: • λ₁, λ₂ (regularization weights), • τ (saliency threshold), and • noise schedule parameters (corruption magnitude or rate). • Could the authors provide sensitivity curves or variance estimates showing GUIDE’s stability with respect to these parameters? • This would be particularly useful to assess GUIDE’s reliability as a “plug-and-play” post-hoc method.
> >
> > A: We thank the reviewer for raising this point. In the current version, we already provide a sensitivity study for the main GUIDE-specific parameters. Table 6 varies the number of curriculum stages T, Table 7 varies the corruption rate γ, and Table 8 varies the saliency coverage threshold η. Across all these settings, in-distribution accuracy remains essentially unchanged and coverage/robustness vary smoothly, with a broad plateau around our default configuration, which we believe is encouraging for practical use. The remaining weights in the evidential loss (e.g., the KL regulariser coefficient) are taken directly from prior EDL work and kept fixed across all datasets; empirically we did not observe brittle behaviour when changing them in preliminary sweeps.
> >
> > > Were all models trained or calibrated using identical datasets and computational budgets? • Since GUIDE is post-hoc, how is its runtime and memory footprint compared to ensemble-based methods? • Including a table summarizing computational cost vs. performance could clarify GUIDE’s real-world efficiency.
> >
> > A: All baselines were trained or calibrated strictly on the same ID datasets using identical splits and comparable computational budgets, which we detail in Appendix C. Since GUIDE is fully post-hoc and does not retrain the backbone, its runtime and memory footprint are substantially lower than ensemble-based approaches, which require multiple forward passes and multiple stored models. GUIDE performs a single saliency pass per sample during calibration and a single forward pass through the meta-model at test time, making its inference cost comparable to other lightweight post-hoc methods such as Whitebox or EMM at inference. We will look to perform a computational analysis as future work.

---

> > > ### Author Response · Authors · 2025-11-21
> > >
> > > > On Theoretical Result (Theorem 1 – Fisher Information Retention) • Theorem 1 provides a bound on retained Fisher information under saliency selection. • Could the authors elaborate on the assumptions underlying this bound (e.g., independence or linearity of selected features)? • Is this guarantee empirical (observed in finite data) or asymptotic (as dataset → ∞)? • How tight is the bound in practice — do the authors have empirical measurements correlating Fisher information loss with performance degradation?
> > >
> > > A: We thank the reviewer for the question. Theorem 1 is intended as a simplified justification for our saliency-based layer selection, not as a full finite-sample or causal theory of uncertainty. The bound relies on standard local linearisation of the pretrained network and on constructing orthogonal “innovation’’ features so that Fisher information decomposes across depths; this does not assume independence, only orthogonality after projection. The factors $\kappa$ and $\rho$ explicitly capture the two modelling assumptions: that LRP saliency is reasonably aligned with per-layer Fisher mass and that the linear projections used by the meta-model are not pathologically lossy. In practice, the result functions as an information-preservation sanity check. Figure 8 in our appendix provides an empirical approximation of this behaviour, showing that the cumulative information retained by our selected layers closely matches the target $\eta$.
> > >
> > > > On the Relationship Between GUIDE and Calibration Methods • GUIDE seems related in spirit to post-hoc calibration (e.g., Temperature Scaling, Platt Scaling, Isotonic Regression) but introduces a learning component. • How does GUIDE differ conceptually and practically from these calibration techniques in terms of the underlying uncertainty model (Dirichlet vs. softmax temperature)? • Could the authors provide a brief comparison or unified framework situating GUIDE among existing calibration approaches?
> > >
> > > A: While GUIDE operates in a post-hoc setting, its conceptual and practical goals differ from those of classical calibration techniques such as Temperature Scaling, Platt Scaling, or Isotonic Regression. Calibration methods operate purely at the logit/output level, reshaping softmax probabilities to better align with empirical frequencies on ID validation data. They do not modify representations, do not inspect internal features, and do not learn uncertainty behaviour under distribution shift. In contrast, GUIDE explicitly learns a Dirichlet-based uncertainty model that captures both confidence (total evidence) and epistemic uncertainty (evidence dispersion), enabling uncertainty-aware behaviour beyond calibration.
> > >
> > > GUIDE uses saliency-informed intermediate features and a structured noise curriculum to teach the meta-model when to reduce evidence as salient information is perturbed. This allows GUIDE to respond meaningfully to OOD or adversarial inputs, settings where traditional calibration methods are known to remain overconfident because they only adjust output temperatures on ID data.
> > >
> > > Under a unified view of post-hoc methods, calibration techniques can be interpreted as output-level transformations, whereas GUIDE is a representation-level meta-model that learns uncertainty dynamics from structured perturbations. We will add a short paragraph in the Related Work section clarifying this distinction and situating GUIDE within this broader landscape.
> > >
> > > > On Scalability to Modern Architectures • Have the authors tested GUIDE on transformer-based architectures (e.g., ViT, DeiT) or multimodal models? • If not, do they anticipate challenges due to the saliency extraction step or computational scaling? • This would be important to determine whether GUIDE generalizes beyond CNN-based settings, which are becoming less central in current ICLR research.
> > >
> > > A: Our current experiments already evaluate GUIDE across a diverse set of pretrained architectures, including LeNet, ResNet, and SENet (Squeeze-and-Excitation) backbones. These demonstrate that GUIDE generalises beyond simple feedforward or vanilla CNN structures. The consistent performance across these heterogeneous architectures indicates that GUIDE is not tied to a particular network design. While we have not yet evaluated GUIDE on transformer-based models such as ViT or DeiT, we do not anticipate fundamental challenges. Evidently, we will look into these architectures as future work.

---

> > > > ### Author Response · Authors · 2025-11-21
> > > >
> > > > > On Dataset Diversity and Real-World Scenarios • The experiments are limited to canonical image datasets (MNIST, CIFAR, SVHN, Oxford Flowers). • Could the authors discuss potential extensions or ongoing work on realistic, domain-shifted, or corrupted datasets (e.g., CIFAR-C, ImageNet-C, or medical images)? • This would strengthen the claim of GUIDE’s utility in safety-critical applications.
> > > >
> > > > A: We agree that natural distribution-shift benchmarks such as CIFAR-10-C or ImageNet-C would be valuable additions. While we do not include corruption benchmarks in this version, our evaluation already spans a wide range of natural domain shifts across MNIST, FashionMNIST, KMNIST, EMNIST, CIFAR-10, CIFAR-100, SVHN, Oxford Flowers, and DeepWeeds. These dataset changes induce substantial appearance and distributional variation beyond adversarial perturbations. We will look to compare against these additional datasets as a part of our future work.
> > > >
> > > > > On Interpretability and Human Alignment • Since GUIDE integrates saliency-driven learning, it implicitly aligns with explainability goals. • Have the authors considered whether the generated saliency maps or uncertainty overlays are interpretable by human users (e.g., can humans trust GUIDE’s “when uncertain” behavior)? • A small user study or interpretability evaluation could add practical value to this aspect.
> > > >
> > > > A: GUIDE indeed incorporates saliency-driven components, and as a result, it inherits a degree of interpretability from the LRP-based relevance maps used during calibration. In our internal qualitative analysis (e.g., for airplane and animal classes), the extracted relevance maps consistently highlight semantically meaningful regions such as object contours and discriminative parts, rather than background artifacts. Similarly, the selected layers typically correspond to mid- to high-level semantic features, providing an interpretable summary of “where” the pretrained model stores task-relevant information. While GUIDE’s primary contribution is uncertainty estimation rather than user-facing interpretability, we agree that visualising saliency maps or uncertainty overlays can offer complementary insights into when and why the model becomes uncertain. We can add qualitative examples in the camera-ready version (if accepted) to illustrate this behaviour.
> > > >
> > > > > On Open-Source and Reproducibility • The paper claims that code will be made available. • Could the authors clarify the current status of code release and whether pretrained models, saliency scripts, and evaluation setups will be publicly accessible? • Reproducibility is especially important for a post-hoc framework advertised as “lightweight and widely applicable.”
> > > >
> > > > A: We thank the reviewer for stressing this. We provide an anonymised open-source repository link in the main paper (line 296 - footnote 2 on page 6), containing scripts to reproduce all experiments. Upon acceptance, we will de-anonymise the repository to ensure it is fully reproducible.

---

### Meta-Review · Area_Chair_6Ab8 · 2026-01-05

**Summary:**

The paper proposes GUIDE, a post-hoc framework for uncertainty quantification (UQ) in deep neural networks. The method attaches a lightweight evidential meta-model to a frozen pretrained backbone. It uniquely combines Layer-wise Relevance Propagation (LRP) to identify salient features and a noise-driven curriculum to teach the meta-model when to express uncertainty.

**Reviewer Concerns:**

The method is merely a combination of existing techniques (EDL, saliency, curriculum learning). Confusion regarding the definition of "post-hoc" (vs. end-to-end EDL) and the limitation to CNN-based classifiers (vs. transformers/foundation models). Why LRP was chosen over other saliency methods (e.g., Grad-CAM) and why stronger attacks (AutoAttack) were not used.

**Reviewer Scores:**

Reviewer 7wVu suggested that using PGD/FGSM is outdated and AutoAttack should be the standard. The authors defended their choice to ensure comparability with existing post-hoc baselines but acknowledged AutoAttack as a necessary future addition. Reviewer A2To felt the concept of "learning when to be uncertain" lacked a formal causal or probabilistic derivation, remaining largely intuitive. While the authors provided intuition on the curriculum design, a rigorous theoretical formalism for this specific learning dynamic remains absent.

---

### Decision · Program_Chairs · 2026-01-26

Reject